# EchoVLM: Measurement-Grounded Multimodal Learning for Echocardiography

## Abstract

Echocardiography is the most widely used imaging modality in cardiology, yet its interpretation remains labor-intensive and inherently multimodal, which requires view recognition, quantitative measurements, qualitative assessments, and guideline-based reasoning. While recent vision–language models (VLMs) have achieved broad success in natural images and certain medical domains, their potential in echocardiography has been limited by the lack of large-scale, clinically grounded image–text datasets and the absence of measurement-based reasoning central to echo interpretation. We introduce EchoGround-MIMIC, the first measurement-grounded multimodal echocardiography dataset, comprising 19,065 image–text pairs from 1,572 patients with standardized views, structured measurements, measurement-grounded captions, and guideline-derived disease labels. Building on this resource, we propose EchoVLM, a vision–language model that incorporates two novel pretraining objectives: (i) a view-informed contrastive loss that encodes the view-dependent structure of echocardiographic imaging, and (ii) a negation-aware contrastive loss that distinguishes clinically critical negative from positive findings. Across five types of clinical applications with 36 tasks spanning multimodal disease classification, image–text retrieval, view classification, chamber segmentation, and landmark detection, EchoVLM achieves state-of-the-art performance (86.5% AUC in zero-shot disease classification and 95.1% accuracy in view classification). We demonstrate that clinically grounded multimodal pretraining yields transferable visual representations and establish EchoVLM as foundation model for end-to-end echocardiography interpretation. We will release EchoGround-MIMIC and data curation code, enabling reproducibility and further research in multimodal echocardiography interpretation.

## 1 Introduction

Echocardiography (cardiac ultrasound) is the most widely used imaging technique in cardiology due to its safety (non-invasive, radiation-free), portability, and low cost. Given its high volume usage, clinicians are routinely tasked with interpreting large numbers of studies within limited time constraints. The clinical workflow for interpreting an echo study is inherently multi-modal and complex. In a standard echo exam, clinicians first identify standardized views of the heart and extract quantitative measurements (*e.g.*, ejection fraction, chamber dimensions, valve gradients). Following guideline-based criteria, these measurements are combined with qualitative assessments (*e.g.*, morphology) to determine disease findings. Finally, these findings are transcribed into a narrative report using natural language to document diagnoses in medical records. This highlights the need for automated AI systems that can support end-to-end, multi-modal echo image analysis.

Despite rapid progress in medical AI, most echo models remain task-specific—strong on single vision tasks but difficult to transfer and insensitive to the cross-modal structure of clinical reading (Leclerc et al., 2019b; Ouyang et al., 2020a). Foundation models (FMs) trained on large, weakly supervised corpora generalize broadly across downstream tasks (Radford et al., 2021; Zhang et al., 2024a; Siméoni et al., 2025), but in echocardiography the landscape is fragmented: recent vision-only FMs (e.g., EchoApex (Amadou et al., 2024), EchoFM (Kim et al., 2024)) lack language and measurement context, while echo VLMs (e.g., EchoCLIP (Christensen et al., 2023)) align images with reports without grounding captions in quantitative measurements or guideline logic. A central bottleneck is absence of *measurement-grounded* image–text supervision tailored to echo workflow.

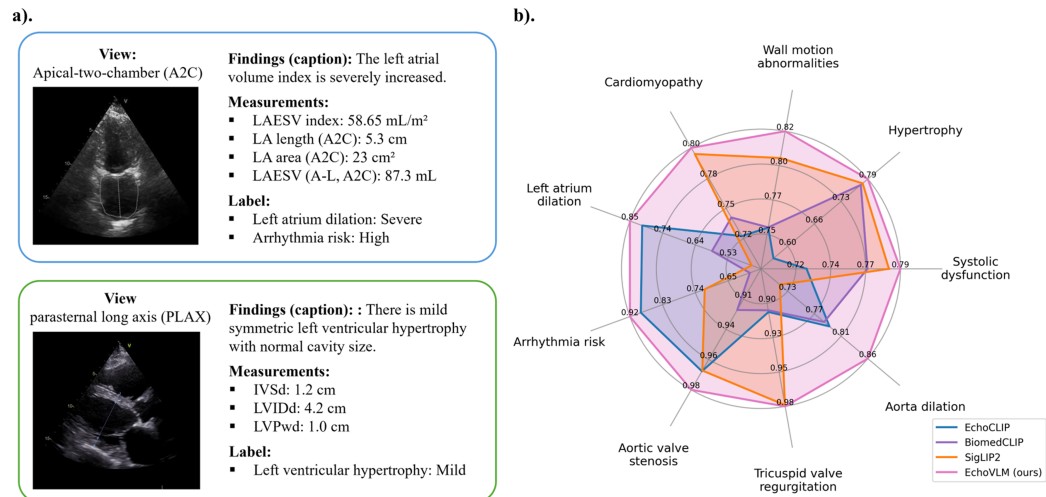

Figure 1: (a) EchoGround-MIMIC examples with standardized view labels (A2C/A4C), OCR-extracted measurements, measurement-grounded captions, and guideline-aligned disease labels. (b) Zero-shot disease classification AUC by category (radar plot): EchoVLM consistently outperforms EchoCLIP, BiomedCLIP, and SigLIP2 (higher is better).

In this work, we introduce *EchoGround-MIMIC*, the first measurement-grounded multimodal dataset for echo, and *EchoVLM*, a vision–language model that encodes clinical priors essential for faithful interpretation. EchoGround-MIMIC contains 19,065 image–text pairs from 1,572 patients, each linked to an ASE-standard view label, OCR-extracted structured measurements, LLM-derived measurement-grounded captions, and guideline-aligned disease labels. The curation mirrors how cardiologists process echo studies: organize by view, distill quantitative machine measurements from images, and extract sentences that explicitly depend on those measurements; labels are checked for consistency against measurements and captions.

Building on this resource, EchoVLM extends CLIP-style image–text alignment with two clinically informed objectives. A *view-informed contrastive loss* enforces intra-view coherence and inter-view separation, reflecting the view-dependent nature of echo acquisition. A *negation-aware contrastive loss* contrasts original captions with counterfactual negated variants (e.g., "no regurgitation" vs. "mild regurgitation"), improving discrimination of clinically critical negatives. For quantitative statement such as EF is 45%, we map the value to its clinical interpretation and negate that interpretation, e.g. "no systolic dysfunction". Together these objectives inject echo-specific priors into multimodal pretraining.

In summary, our contributions are

1. **Measurement-grounded echocardiography dataset.** We introduce *EchoGround-MIMIC*, pairing echo images with standardized views, structured measurements, measurement-grounded captions, and guideline-derived disease labels, enabling training and rigorous evaluation of clinically faithful VLMs. We will release the dataset together with the pre-processing code.

2. **Clinically informed multimodal pretraining.** We propose *EchoVLM* with view-informed and negation-aware contrastive objectives on top of CLIP, explicitly encoding view structure and negative findings central to echo diagnosis.

3. **Comprehensive validation.** We systematically evaluate EchoVLM on 5 different clinical applications with 36 clinical tasks. Across multimodal and vision-only tasks, EchoVLM achieves state-of-the-art performance: zero-shot disease classification (AUC 86.5%, precision 34.2%), best Top-5/Top-10 image–text retrieval recall, 95.1% accuracy on downstream view classification (surpassing the strongest vision FM), strong interactive segmentation and competitive landmark detection on public datasets consisting of over 10K annotated images. Our ablations confirm complementary gains from proposed objectives.

## 2 RELATED WORK

**Medical vision–language models.** Large-scale image–text pretraining has enabled generalizable representation learning across tasks and domains. CLIP popularized contrastive alignment between vision and language, demonstrating strong zero-shot transfer from large scale image–text pairs and catalyzing the modern VLM paradigm ((Radford et al., 2021)). Subsequent work emphasized the role of data curation (*e.g.,* MetaCLIP (Xu et al., 2023) and MetaCLIP 2 (Chuang et al., 2025)), showing that transparent, balanced selection from web corpora can rival or surpass original CLIP data across benchmarks. Beyond general-domain VLMs, the biomedical community has trained domain-adapted models on scientific figures and clinical images. BiomedCLIP pretrains on large PMC-scale figure–caption pairs and reports strong transfer across radiology and biomedical tasks ((Zhang et al., 2023)). Complementary efforts (*e.g.,*, PMC-CLIP (Lin et al., 2023)) curate millions of high-fidelity biomedical image–text pairs to improve retrieval, classification, and VQA. This line of work establishes the viability of contrastive VLMs in clinical settings but typically lacks explicit modeling of measurement-grounded semantics crucial for echocardiography.

**Foundation models for echocardiography.** Self-supervised vision pretraining has matured through contrastive and masked reconstruction objectives, yielding robust visual features that transfer with minimal labels ((Chen et al., 2020; He et al., 2022; Caron et al., 2021; Oquab et al., 2024; Siméoni et al., 2025)). Recent works have begun to specialize foundation modeling to echo. For vision-only FMs, EchoApex ((Amadou et al., 2024)) pretrains an in-domain visual backbone on ∼20M echo images spanning transthoracic (TTE), transesophageal (TEE), and intracardiac echocardiography (ICE), across B-mode, Doppler, and 3D acquisitions, then adapts with task-specific heads to vision tasks . EchoFM ((Kim et al., 2024)) targets video representation learning with spatio-temporal masking and periodic-driven contrastive learning, pretraining on ∼290k echo videos and transferring to four downstream tasks. These models underscore the value of large in-domain pretraining but treat language and measurement semantics only indirectly. EchoCLIP ((Christensen et al., 2023)) scales echo VLM training to over one million video–text pairs mined from clinical reports, enabling zero-shot assessment and retrieval. EchoPrime ((Zhang et al., 2024b)) pushes multi-*video* (study-level) learning further: it uses a view classifier and view-aware anatomic attention to aggregate across standardized views, training on 12M video–report pairs and achieving improved performance across diverse form/function benchmarks. While these VLMs exploit report text, their captions largely reflect free-text narratives; quantitative measurements—central to guideline-based echo diagnosis—are not explicitly modeled, leaving negation and threshold-based criteria underrepresented.

## 3 METHOD

### 3.1 DATA CURATION

**Source data** EchoGround-MIMIC is curated from the publicly available MIMIC-IV-ECHO and MIMIC-IV-Note modules (Goldberger et al. (2000); Johnson et al. (2023; 2024)), collected at Beth Israel Deaconess Medical Center. MIMIC-IV-ECHO comprises over 500,000 echocardiogram videos from 7,243 studies across 4,579 patients (2017–2019), each containing embedded machine measurements. MIMIC-IV-Note includes 331,794 de-identified discharge summaries from 145,915 patients, processed under HIPAA Safe Harbor protocols. We identify echocardiography reports within this corpus via string matching and link them to imaging studies using shared identifiers and study timestamps.

**View and measurement extraction** We structure each study to mirror clinical reading: a pretrained classifier assigns standardized views, followed by quantitative measurement extraction. Views are categorized using the American Society of Echocardiography (ASE)–defined classes (Mitchell et al. (2019)). To extract quantitative parameters, we crop the clinically annotated measurement overlays from the image and transcribe them into structured JSON using Qwen2.5-VL-72B (Qwen (2025); Wang et al. (2024)). The resulting fields include chamber dimensions, transvalvular gradients, and Doppler ratios.(Fig. 2a). Extraction fidelity was verified by manual review.

**Extracting measurement-grounded captions** Echo reports are retrieved from discharge summaries and paired with structured measurements. We prompt a state-of-the-art LLM (Qwen2.5-

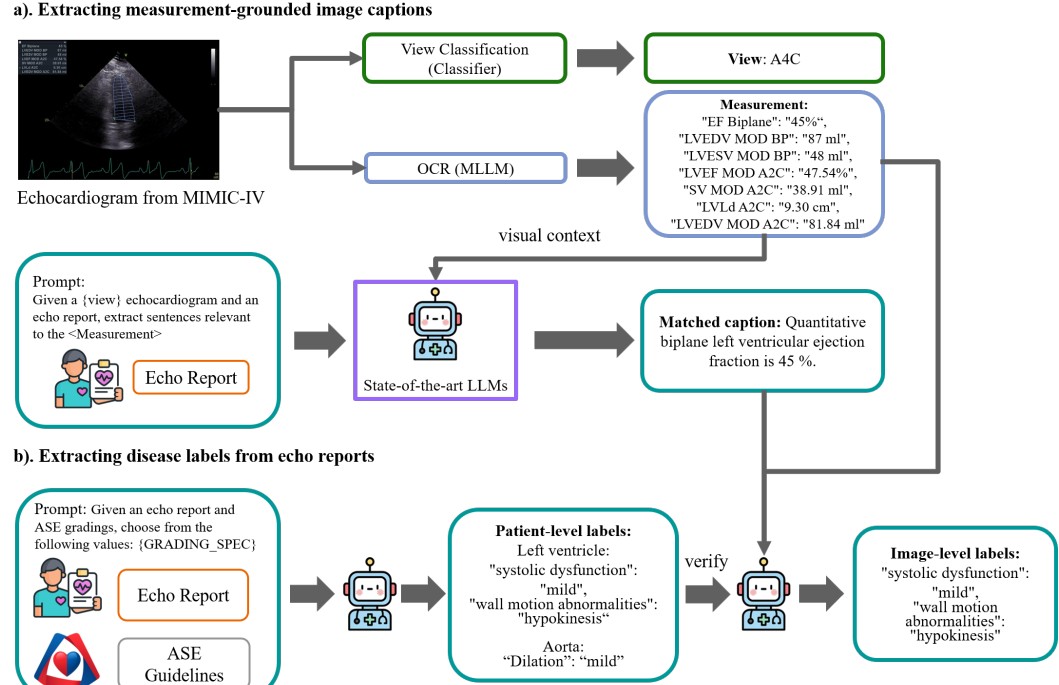

Figure 2: Data curation for EchoGround-MIMIC. (a) For each MIMIC-IV echocardiogram, we perform view classification, OCR the embedded machine measurements, and prompt an LLM with the image and report to produce *measurement-grounded* captions. (b) Using the report and ASE grading schema, an LLM assigns patient- and image-level disease labels, followed by a consistency check against the extracted measurements.

Instruct-72B) to extract sentences explicitly referencing or dependent on these measurements, generating measurement-grounded captions (e.g., "Quantitative biplane left ventricular ejection fraction is 45%"). Studies without such grounded sentences are excluded.

**Guideline-based Abnormality Extraction**   To derive patient-level disease labels, we implement a two-stage procedure (Fig. 2b). First, we prompt LLM to extract abnormalities defined by American Society of Echocardiography guidelines, assigning exactly one severity grade per abnormality (none, mild, moderate, severe). Default assignment is normal when no evidence is found. Second, we perform consistency checking by prompting LLM with both the measurements, captions and the patient-level disease labels to verify alignment with guidelines. The model discards any label that conflicts with measurements or captions. This process ensures that the final abnormality annotations reflect a guideline-consistent interpretation of both measurement and textual content. The process is further detailed in appendix A.4.

**EchoGround-MIMIC**   The resulting dataset, EchoGround-MIMIC, comprises 19,065 image–text pairs from 1,572 patients. Each image is annotated with abnormality labels spanning 9 ASE-defined disease categories (Fig. 3, left) graded from normal to severe (see appendix A.5.1, Fig. 8), as well as one of 22 ASE-standard views (Fig. 3, right). In addition, the dataset includes structured measurements and measurement-grounded captions. This unified resource anchors echocardiography interpretation in standardized views, quantitative measurements, and clinical guidelines, supporting the development and evaluation of multimodal models.

## 3.2 PRETRAINING ECHOVLM WITH CLINICALLY INFORMED OBJECTIVES

Having curated a large-scale multi-modal dataset, our goal is to build a vision-language model that captures the clinical priors inherent to echocardiography. We adopt the standard CLIP (Radford et al., 2021) image-text contrastive framework as a baseline and introduce two novel objectives that

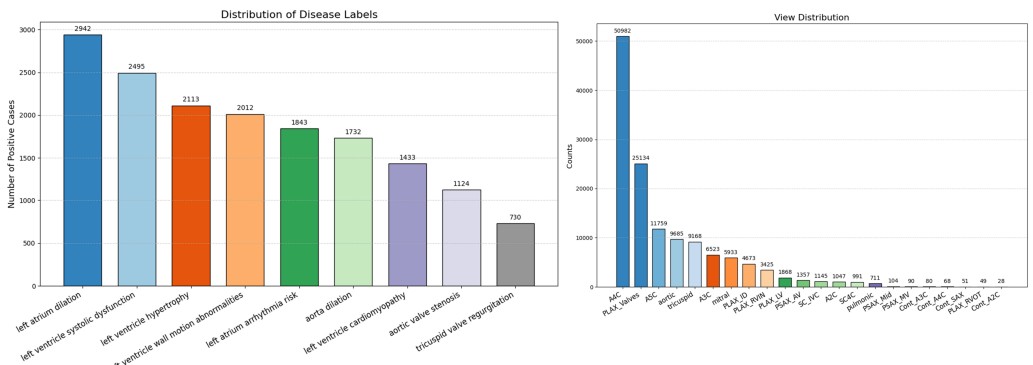

Figure 3: Disease and view label distributions in EchoGround-MIMIC. Left: Binary label distribution for each of the 9 disease categories (labels greater than *mild* are considered positive). Right: Distribution of echocardiographic views in the dataset.

encode such priors: *view-informed contrastive learning* and *negation-aware contrastive learning*. We aim to address two challenges in multi-modal echo interpretation: (i) the view-dependent nature of image content and (ii) the prevalence of clinically critical negations in reports.

**View-informed contrastive learning.** Echocardiographic interpretation is inherently view-dependent, with each standardized acoustic window (e.g., PLAX, A4C) providing distinct anatomical and diagnostic cues. To encode this structure, we introduce a view-informed contrastive loss $\mathcal{L}_{\text{view}}$. For each anchor image, positive examples are drawn from the same view class, while negatives come from different views. This encourages intra-view coherence and inter-view separation in the visual embedding space, mirroring how cardiologists reason within and across views. Let $z_i^{\text{img}} \in \mathbb{R}^d$ be the image embedding, and $v_i$ the discrete view label of image $i$ (e.g., PLAX, A4C). Let the positive set be $P(i) = \{j \mid v_j = v_i, \ j \neq i\}$. The view-informed contrastive loss $L_{\text{view}}$ is

$$\mathcal{L}_{\text{view}} = -\frac{1}{N}\sum_{i=1}^{N}\frac{1}{|P(i)|}\sum_{j\in P(i)}\log\frac{\exp(\tau \cdot z_i^{\text{img}} \cdot z_j^{\text{img}})}{\sum_{k\neq i}\exp(\tau \cdot z_i^{\text{img}} \cdot z_k^{\text{img}})}, \tag{1}$$

where $N$ is the batch size and $\tau$ is a learnable temperature.

**Negation-aware contrastive learning.** Clinical reports frequently use negations (e.g., "no pericardial effusion", "no significant regurgitation"), but standard contrastive training often struggles with understanding negations. To address this, we augment captions by prompting an LLM to rewrite positive findings into their negated forms. We then introduce a negation-aware loss $\mathcal{L}_{\text{neg}}$ that explicitly separates original and negated embeddings:

$$\mathcal{L}_{\text{neg}} = \frac{1}{N}\sum_{i=1}^{N}\text{BCEWithLogits}\big(\tau\, z_i^{\text{txt}} \cdot z_i^{\text{neg}},\, 0\big) \tag{2}$$

where $z_i^{txt}$ and $z_i^{neg}$ denote embeddings of the original and negated captions, respectively. BCEWithLogits($\cdot$) denotes the binary cross-entropy loss with logits. This loss explicitly teaches the model to distinguish negative, normal findings from positive ones, improving the precision of disease detection.

**Total objective** The final pretraining objective $L$ combines the three components:

$$\mathcal{L} = \mathcal{L}_{\text{CLIP}} + \lambda_{\text{view}}\mathcal{L}_{\text{view}} + \lambda_{\text{neg}}\mathcal{L}_{\text{neg}}, \tag{3}$$

where $L_{\text{CLIP}}$ is the CLIP loss and we set $\lambda_{\text{view}}$ to 0.5 and $\lambda_{\text{neg}}$ to 0.1. This design preserves the multi-modal capacity of VLM, while injecting both vision and text supervisions that encode the structured workflow of echocardiography.

**Implementation details**  For EchoVLM, we use ViT-B as the vision backbone and CLIP text encoder as text backbone. During pretraining, we use a global batch size of 512, a learning rate of 1e-4 and weight decay of 0.05. All images are resized to $112 \times 112$. We pretrain for a total of 20 epochs with a linear warmup of 200 steps. Vision encoder is initialized from weights pretrained on an internal echo dataset. Text encoder is initialized from EchoCLIP Christensen et al. (2023). More details in Appendix Table 15.

## 4 RESULTS

### 4.1 CROSS-MODAL RESULTS ON ECHOGROUND-MIMIC

**Evaluation**  We randomly partition EchoGround-MIMIC into training, validation, and testing sets using a ratio of 0.8/0.1/0.1. The final training cohort contains 15,255 image–text pairs, and the test cohort includes 1,911 pairs; the validation set is used for hyperparameter tuning. To evaluate cross-modal performance, we consider two tasks: (1) zero-shot disease classification and (2) image–text retrieval. For zero-shot classification, we construct positive and negative prompts for each disease, compute the VLM's predicted probabilities, and assign the label via argmax. We report standard metrics including precision, recall, and AUC. For retrieval, we compute embeddings for both images and texts in the test set and measure performance by top-5 and top-10 recall. We compare EchoVLM against both domain-specific vision–language models such as EchoCLIP (Christensen et al., 2023), BiomedCLIP (Zhang et al., 2024a) and generalist models such as CLIP (Radford et al., 2021), MetaCLIP (Xu et al., 2023) and SigLIP2 (Tschannen et al., 2025)). All methods are finetuned on EchoGround-MIMIC for fairness.

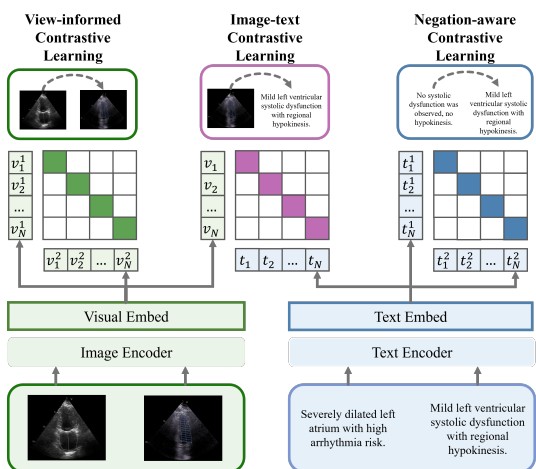

Figure 4: Pretraining objectives in EchoVLM. Left: View-informed contrastive learning enforces intra-view invariance and inter-view separation. Middle: Image–text contrastive learning aligns images with measurement-grounded captions. Right: Negation-aware contrastive learning separates affirmative vs. negated clinical statements (e.g., "no systolic dysfunction" vs. "mild systolic dysfunction").

**EchoVLM outperforms existing VLMs in multi-modal tasks.**  As shown in Table 1, EchoVLM achieves the strongest zero-shot disease classification with an AUC of 86.5% and precision of 34.2%, surpassing the next best model, EchoCLIP, by 7.2% and 6.9%, respectively. On image–text retrieval, EchoVLM also leads with recall of 2.98% at top-5 and 5.70% at top-10, exceeding the strongest baseline by 0.26% and 0.57%. These results demonstrate that incorporating echo-specific priors during pretraining yields transferable representations that generalize robustly across multi-modal echocardiography tasks.

### 4.2 DOWNSTREAM VISION TASKS

**View classification**  We assess the quality of visual representations by finetuning each VLM's vision encoder on a private, multi-vendor TTE dataset collected from six sites (26k videos with 18 ASE-standard views, including contrast). All data are de-identified and converted to grayscale. The dataset is split into 21.5k/2.1k/2.1k videos for training, validation, and testing, respectively. A linear classification head is attached to the pooled visual embeddings, and the encoder is fine-tuned end-to-end (details in Appendix Table 10).

**EchoVLM achieves state-of-the-art performance in view classification.**  EchoVLM achieves the best performance among all evaluated models (Table 2), reaching 95.1% accuracy, 95.3% F1 score, and 95.8% precision. Notably, EchoVLM not only surpasses all VLM baselines but also

| Model | Disease Zeroshot | | | Retrieval | |
|---|---|---|---|---|---|
| | AUC | Precision | Recall | Recall Top-5 | Recall Top-10 |
| EchoCLIP (Christensen et al., 2023) | 79.3 | 27.3 | 75.1 | 1.94% | 4.03% |
| EchoPrime (Vukadinovic et al., 2024) | 82.8 | 29.7 | 93.8 | 1.41% | 2.78% |
| BiomedCLIP (Zhang et al., 2024a) | 77.1 | 20.4 | 77.2 | 2.35% | 5.13% |
| CLIP (B/16) (Radford et al., 2021) | 73.9 | 23.8 | 73.1 | 2.72% | 4.60% |
| SigLIP2 (B/16) (Tschannen et al., 2025) | 78.1 | 22.6 | 86.0 | 1.98% | 4.19% |
| MetaCLIP (B/16) (Xu et al., 2023) | 79.1 | 21.5 | 84.3 | 2.35% | 4.91% |
| **EchoVLM (ours)** | **86.5** | **34.2** | **86.2** | **2.98%** | **5.70%** |

Table 1: Image-text tasks on EchoGround-MIMIC by comparing with existing VLMs. EchoVLM performs strongly in both disease zero-shot and image-text retrieval. **BOLD** means best result.

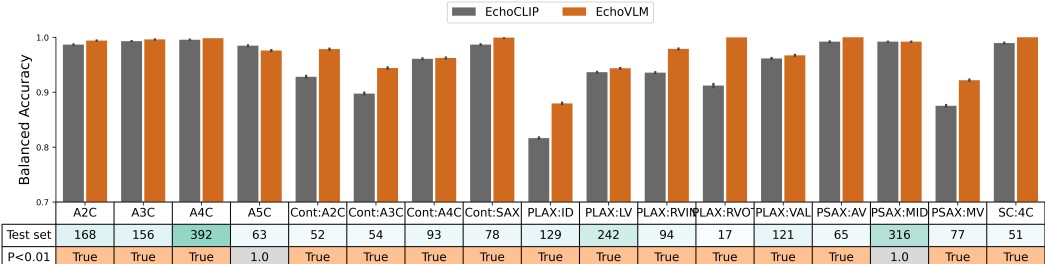

Figure 5: Class-wise view classification (balanced accuracy). EchoVLM vs. EchoCLIP. Row "Test set" reports the number of test images per class; row "$p < 0.01$" marks whether EchoVLM's gain is significant (one-sided $t$-test). EchoVLM achieves higher balanced accuracy on most views.

outperforms the strongest vision foundation model EchoApex by 0.9% in precision. We also provide class-wise analysis in Figure 11. EchoVLM consistently achieves higher balanced accuracy than EchoCLIP across most ASE-standard views, with improvements that are statistically significant for the majority of classes ($p < 0.01$, one-sided $t$-test). These results highlight that our pretraining strategy yields transferable and clinically meaningful visual features, extending beyond multi-modal alignment to purely vision-based tasks.[1]

**Interactive segmentation**  We adapt EchoVLM for chamber segmentation tasks by attaching a prompt-based (box) encoder-decoder module following SAM (Kirillov et al., 2023). Training and evaluation is conducted on three public benchmarks: EchoNet-Dynamic (left ventricle masks in A4C views) (Ouyang et al., 2020b), EchoNet-Pediatric (left ventricle masks in A4C, PSAX views) (Reddy et al., 2023) and CAMUS (left ventricle and atrium masks in A2C views) (Leclerc et al., 2019a). We report Dice similarity coefficient (DSC) and compare with task-specific baselines (U-Net (Ronneberger et al., 2015) or Deeplabv3 (Chen et al., 2017)), MedSAM (Ma et al., 2024) and the vision FM EchoApex (Amadou et al., 2024).

**EchoVLM outperforms tasks-specialists and achieves similar performance as vision FM.** EchoVLM attains the best DSC on EchoNet-Dynamic (93.1%) and EchoNet-Pediatric-A4C (92.4%), and ties EchoApex on EchoNet-Pediatric-PSAX (93.0%) (Table 3). On the CAMUS dataset, EchoVLM matches EchoApex for left ventricular segmentation (93.8%) and achieves competitive performance for left atrial segmentation (90.2%). Visualization of segmentation results on CAMUS using EchoVLM is shown in Figure 6. These results indicate that our pretraining maintains transferable local features for segmentation across datasets.

**Landmark detection**  We further evaluate EchoVLM on landmark detection using the public EchoNet-LVH dataset (Duffy et al., 2022), which benchmarks left ventricular hypertrophy (LVH) assessment from PLAX echocardiographic frames. The task requires predicting landmark coordinates

[1]As EchoPrime is a video-based model, the results are evaluated based on using 16 consective frames rather than single image from the echo video.

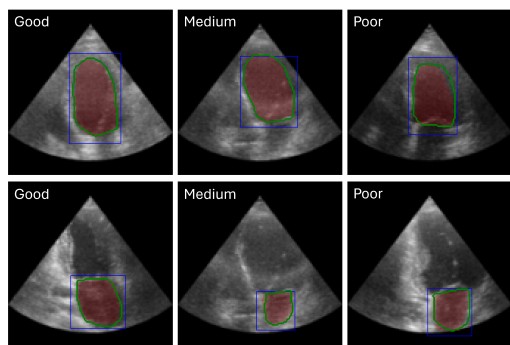

Figure 6: EchoVLM segmentation on CAMUS dataset. Image quality labels: Good, Medium, Poor. Red = prediction, Green = annotation.

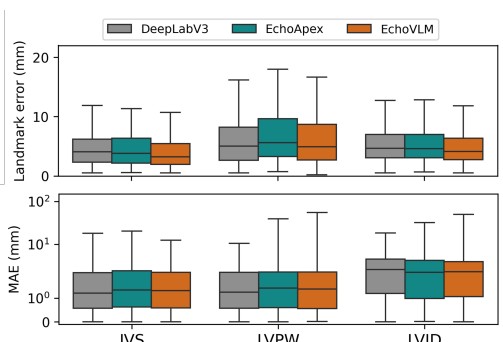

Figure 7: EchoVLM landmark detection on EchoNet-LVH dataset. Boxplots present error distributions (mm) for IVS, LVPW, and LVID.

| Model | Accuracy | F1 | Precision |
|---|---|---|---|
| *Vision foundation model* | | | |
| EchoApex (Amadou et al., 2024) | 94.8 | 94.7 | 94.9 |
| *VLMs* | | | |
| EchoCLIP (Christensen et al., 2023) | 90.9 | 91.6 | 93.2 |
| EchoPrime(Vukadinovic et al., 2024) | 93.8 | 94.9 | 93.2 |
| BiomedCLIP (Zhang et al., 2024a) | 92.6 | 92.3 | 92.9 |
| CLIP (B/16) (Radford et al., 2021) | 90.5 | 87.8 | 86.7 |
| SigLIP2 (B/16) (Tschannen et al., 2025) | 89.7 | 89.1 | 89.4 |
| MetaCLIP (B/16) (Xu et al., 2023) | 87.4 | 86.1 | 86.7 |
| **EchoVLM (ours)** | **95.1** | **95.3** | **95.8** |

Table 2: Performance of VLMs and the vision FM on downstream view classification. EchoVLM achieves the highest accuracy, F1, and precision.

for the interventricular septum (IVS), left ventricular internal dimension (LVID), and left ventricular posterior wall (LVPW), from which wall thicknesses are derived. For network architecture, we attach a UNETR style decoder to EchoVLM Hatamizadeh et al. (2022). Evaluation metrics include the mean absolute error (MAE) of the derived measurements (mm) and the average landmark error (Average L.E), defined as the Euclidean (L2) distance between predicted and ground-truth landmark coordinates (mm). The dataset is split into 20,254/2,275/683 frames for training, validation, and testing, respectively. We compare with a task-specific model DeepLabV3 (Chen et al., 2017) and foundation model EchoApex. As shown in Table 4, EchoVLM achieves competitive performance, outperforming both baselines on IVS and LVID in both metrics.

## 4.3 ABLATION STUDY ON DATA CURATION

We conduct an ablation on the importance of data curation by comparing performance between pretraining EchoVLM using the raw echocardiography report and using the curated grounded captions. To isolate the effect of text data, we did not use additional supervision strategies such as view contrastive or text negation loss. We used the same training and evaluation split and pretraining setup as in our main experiment. As in Table 5, training on raw reports leads to substantially worse multimodal alignment. In zero-shot disease classification, AUC drops from 79.6 to 54.4, and precision drops from 29.0 to 12.5. We suspect the reason for such decline is that raw clinical reports contain large amounts of non-specific information that impedes image-text alignment. Therefore, we leverage OCR-extracted measurements and ASE guideline to curate both visually and clinically-grounded captions. We believe that our curated EchoGround-MIMIC provides substantial and necessary benefit for training a clinically reliable VLM.

| Model | EchoNet-Dynamic | EchoNet-Pediatric | | CAMUS | |
|---|---|---|---|---|---|
| | | A4C | PSAX | LV | LA |
| EchoApex (Amadou et al., 2024) | 92.8 | 92.1 | 93.0 | **93.8** | **90.8** |
| Specialist (Ronneberger et al., 2015) | 91.5 | 89.1 | 89.6 | 92.8 | 90.4 |
| MedSAM (Ma et al., 2024) | 86.5 | 87.2 | 88.0 | 87.2 | 80.3 |
| **EchoVLM (ours)** | **93.1** | **92.4** | **93.1** | **93.8** | 90.2 |

Table 3: Downstream segmentation performance (DSC (%)) on public datasets: EchoNet-Dynamic, EchoNet-Pediatric and CAMUS. EchoVLM consistently outperforms task-specific models (U-Net, MedSAM) and performs on par with vision FM model.

| Model | IVS | | LVID | | LVPW | |
|---|---|---|---|---|---|---|
| | Average L.E | MAE | Average L.E | MAE | Average L.E | MAE |
| DeepLabV3 (Chen et al., 2017) | 4.68 | 1.77 | 5.35 | 3.23 | **5.94** | **1.56** |
| EchoApex (ViT-B) (Amadou et al., 2024) | 4.73 | 2.03 | 5.49 | 3.06 | 6.99 | 1.81 |
| **EchoVLM (ours)** | **4.15** | **1.70** | **5.30** | **3.04** | 6.56 | 1.67 |

Table 4: Downstream landmark detection results on EchoNet-LVH. EchoVLM surpasses the task-specific model DeepLabV3 and the vision FM EchoApex on IVS and LVID. Lower is better.

| Data Source | AUC | Precision | Recall | Recall@5 | Recall@10 |
|---|---|---|---|---|---|
| Raw Reports | 54.43 | 12.47 | 66.62 | 0.31 | 0.78 |
| Curated Captions | 79.60 | 29.00 | 73.60 | 2.30 | 4.33 |

Table 5: Ablation study on curated captions.

## 4.4 ABLATION STUDY ON PRETRAINING OBJECTIVES

We evaluate impact of each proposed objective on (i) cross-modal zero-shot disease classification and (ii) vision-only view classification (Table 6). We pretrain VLM with each ablated loss configuration on EchoGround-MIMIC and fine-tune vision encoder on TTE dataset for view classification.

**View-informed loss**   Incorporating $\mathcal{L}_{\text{view}}$ improves view recognition accuracy considerably by 1.1%. On zero-shot disease detection, we also find considerable gains in precision (+4.5%), recall (+4.5%) and AUC (+1.1%).

**Negation-aware loss**   Adding $\mathcal{L}_{\text{neg}}$ primarily benefits text–image alignment. Zero-shot disease classification improves by 2.6% in AUC and 2.5% in recall. We also found modest improvements to view classification, indicating indirect benefits for the vision encoder.

**Combined objectives**   When both objectives are applied alongside $\mathcal{L}_{\text{CLIP}}$, EchoVLM achieves the best performance across all metrics: 86.5% AUC, 34.2% precision, and 86.2% recall for disease zero-shot classification, and 95.1% accuracy for view classification. These results demonstrate that $\mathcal{L}_{\text{view}}$ and $\mathcal{L}_{\text{neg}}$ are complementary and produce meaningful representations in the multi-modal embedding space, without compromising vision features.

| Pretrain losses | Disease Zeroshot | | | View Classification | | |
|---|---|---|---|---|---|---|
| | AUC | Precision | Recall | Accuracy | F1 | Precision |
| $L_{\text{CLIP}}$ | 79.6 | 29.0 | 73.6 | 92.9 | 93.1 | 93.9 |
| $L_{\text{CLIP}} + L_{\text{View}}$ | 80.7 | 33.5 | 78.1 | 94.0 | 94.4 | 95.2 |
| $L_{\text{CLIP}} + L_{\text{Negation}}$ | 83.3 | 30.5 | 81.6 | 94.4 | 94.4 | 94.9 |
| $L_{\text{CLIP}} + L_{\text{View}} + L_{\text{Negation}}$ | **86.5** | **34.2** | **86.2** | **95.1** | **95.3** | **95.8** |

Table 6: Ablation study on pretraining objective. Cross-modal and vision-only results are reported.

**Loss ratio sensitivity** We further examine the effects of weighting $\lambda_{\text{view}}$ and $\lambda_{\text{neg}}$ on zero-shot disease classification (Table 7). Equal weighting yields modest AUC (79.6%) and precision (29.0%). Reducing the view-informed loss ($\lambda_{\text{view}} = 0.25$, $\lambda_{\text{neg}} = 0.5$) increases recall to 87.2% but at the expense of specificity. Finally, reducing text negation loss ($\lambda_{\text{view}} = 0.5$, $\lambda_{\text{neg}} = 0.1$) achieves the best performance with the highest AUC (83.9%) and precision (32.2%) while maintaining strong recall (83.4%).

| $\lambda_{\text{view}}$ | $\lambda_{\text{neg}}$ | AUC | Precision | Recall |
|---|---|---|---|---|
| 1.0 | 1.0 | 79.6 | 29.0 | 73.6 |
| 0.25 | 0.5 | 81.8 | 28.0 | **87.2** |
| 0.5 | 0.5 | 78.1 | 26.3 | 81.3 |
| 0.5 | 0.1 | **83.9** | **32.2** | 83.4 |

Table 7: Ablation study with vary loss ratios.

## 4.5 INFERENCE EFFICIENCY IN CLINICAL WORKFLOWS

To assess the suitability of EchoVLM for real-time or near–real-time clinical environments, we performed a profiling study on an NVIDIA A100 GPU to quantify end-to-end inference latency and peak memory usage across all downstream tasks. EchoVLM uses lightweight task-specific decoders (e.g., linear heads, SAM-based segmentation heads, and compact UNet architectures), enabling fast inference and modest memory consumption across applications.

| Task | Model Size | Decoder Type | Latency | Memory |
|---|---|---|---|---|
| Disease Classification | 393.9M | CLIP | 251 ms | 4.19 GB |
| View Classification | 86.6M | Linear | 5 ms | 0.83 GB |
| Segmentation | 105.8M | SAM | 9.5 ms | 2.15 GB |
| Landmark Detection | 99.4M | UNETR | 15.3 ms | 2.49 GB |

Table 8: Inference latency and peak GPU memory usage for EchoVLM across downstream tasks, measured on an NVIDIA A100 GPU.

As shown in Table 8, EchoVLM achieves sub-second latency for all tasks, with most operations completing within 5–20 ms, and peak memory usage remaining below 5 GB. Such efficiency makes the model well-suited for both real-time image interpretation and post-exam analysis in routine echocardiography workflows.

## 5 CONCLUSION

This work introduces EchoGround-MIMIC, a measurement-grounded multimodal dataset for echocardiography, and EchoVLM, a vision–language model that encodes clinical priors via view-informed and negation-aware objectives. EchoVLM achieves state-of-the-art results across multimodal tasks, while transferring strongly to vision-only tasks. Future work will focus on scaling our approach to multi-institutional data with diverse acquisitions and temporal modeling. Our approach also has a few limitations. First, EchoGround-MIMIC is derived from a single healthcare system and time window, which may limit demographic and protocol diversity. Second, quantitative values are obtained via OCR from overlays. Despite manual checks, parsing errors may introduce label noise. Third, measurement-grounded captions and guideline labels are generated with large language models. While consistency checks reduce obvious errors, such supervision cannot fully replace expert adjudication and may propagate biases. For these reasons, we do not recommend direct clinical use of EchoGround-MIMIC labels or EchoVLM outputs for diagnosis without expert validation.

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

# A APPENDIX: TABLES AND FIGURES

CONTENTS

## A.1 USAGE OF LLMS

Large Language Models were used in the data curation pipeline but not in the ideation of this manuscript. Specifically:

- **Measurement parsing and caption generation.** We used Qwen2.5-VL-72B to transcribe embedded measurement overlays from echocardiography images into structured JSON, and Qwen2.5-Instruct-72B to extract measurement-grounded captions from associated clinical reports.
- **Guideline-based disease labeling.** We prompted an LLM to assign disease severity grades (none, mild, moderate, severe) according to American Society of Echocardiography (ASE) guidelines. A secondary LLM pass was used to verify label–measurement consistency.
- **Text negation.** We employed an LLM to rewrite affirmative captions into their negated forms (e.g., "mild regurgitation" → "no regurgitation") for use in the negation-aware contrastive loss.

All outputs generated by LLMs were subjected to automated consistency checks and sub-sampled manual review to minimize errors and biases. LLMs did not contribute to study design or analysis. We used LLM to polish the writing of the paper.

## A.2 ETHICS STATEMENTS

This work uses de-identified echocardiography data from two sources. First, we curate EchoGround-MIMIC from the publicly available MIMIC-IV-Echo and MIMIC-IV-Note modules, which are released under HIPAA Safe Harbor protocols with prior IRB approval at the source institution. Second, we evaluate model transferability on a private, multi-vendor institutional dataset collected across six

clinical sites. All private data were fully de-identified before use, and no additional patient recruitment or human subject experimentation was conducted by the authors. Our experiments focus on developing general-purpose vision–language models for echocardiography interpretation and do not provide diagnostic outputs intended for direct clinical use. While our models achieve strong performance, outputs may still contain errors and should not replace expert judgment. Dataset release and code will comply with the terms of use of the underlying MIMIC databases; no identifiable or sensitive information from private institutional datasets will be shared. We believe our study complies with the ICLR Code of Ethics, including fairness, transparency, and responsible use of medical data.

## A.3 REPRODUCIBILITY STATEMENT

We have taken several steps to ensure reproducibility of our work. Full details of data curation (view classification, OCR parsing, caption generation, and label extraction) are provided in Section 3 and Appendix A.4.1. Pretraining and loss formulations are explicitly defined in Section 3.2, with pseudocode included in Algorithm 1. Hyperparameters, architectures, and training schedules are described in Section 3.2 and the Appendix. Evaluation protocols for both multimodal and vision-only downstream tasks are detailed in Sections 4.1–4.3, with dataset splits provided in Appendix. We will release code for data preprocessing and the dataset EchoGround-MIMIC. Our goal is to enable independent verification and extension of EchoVLM across multi-modal echocardiography tasks.

## A.4 ADDITIONAL ECHOGROUND-MIMIC DATASET DETAILS

### A.4.1 EXAMPLE PROMPT

In EchoGround-MIMIC, each caption was rigorously verified against OCR-extracted measurements to ensure consistency with quantitative values such as ejection fraction, chamber size, and wall motion. Below we show the full prompt used for verifying captions given measurements.

---

**Prompt**

**TASK:** Read the measurements extracted from an echo image (the *OCR block*) and decide which pre-written *caption* best describes the image.

- Select exactly **one** caption that is most clinically specific (break ties by first appearance).
- **Discard** any caption that conflicts with *any* measurement (e.g., EF %, cavity size, wall motion).
- If no caption is consistent with the OCR block, return an empty array.
- Preserve the exact caption text without modification.

**Output format (must be exact):**

```
OCR block:
{{ocr}}

captions:
{{caption}}

Return ONE of the following and nothing else:

1. {"caption": ["SELECTED CAPTION"]}     (list length = 1)
2. {"caption": []}                       (if no caption fits)
```

---

## A.5 OCR-BASED MEASUREMENT EXTRACTION AND LLM CAPTION VALIDATION

**OCR-based measurement extraction.** We use OCR exclusively to extract measurement name–value pairs from echocardiographic overlays. The initial OCR pass produced 1,232 unique keys,

many representing identical measurements with varied naming conventions (e.g., "AV_Vmax" vs. "AV Vmax") or non-clinical display elements such as gain, velocity scale, or time scale.

To standardize these measurements, we focused on keys appearing more than ten times, yielding 278 candidates. Following clinical reporting practice, we organized all measurements into 11 anatomical categories (LV, LA, RV, RA, MV, TV, AV, PV, SV, pulmonary vein, aorta). Through manual review, these were consolidated into 167 final structured measurements. This curation significantly improves measurement consistency and reduces OCR-related noise.

**Validation of LLM-generated captions.** To ensure the reliability of measurement-grounded captions, we collaborated with a cardiologist to establish clinically appropriate normal ranges for quantitative measurements. We then applied a rule-based consistency check: for each disease category, we compared the LLM-generated diagnostic statement with a rule-based label inferred directly from the numerical value.

Across all disease categories, we observed strong agreement in approximately 87% of cases, typically where values were clearly normal or abnormal. The remaining $\sim$13% involved borderline or subjective clinical ranges. For example, although an LA length of 3.5–5.2 cm is generally normal, a value of 4.9 cm may still be described as "dilated" in practice depending on patient-specific factors such as age or comorbidities. For transparency, we retain these samples but annotate them with a binary "subjective" flag, enabling users to re-filter or reinterpret them as needed.

During manual verification, we removed disease categories where measurement–caption inconsistencies were most prominent—including mitral valve disease, mitral regurgitation, and stroke-volume–related labels. These diagnoses rely on multi-parameter Doppler criteria and hemodynamic context not available in our extracted measurements. Excluding these categories improves dataset reliability and avoids introducing clinically implausible labels.

### A.5.1 DATA DISTRIBUTIONS

Figure 8 further shows that the curated dataset captures fine-grained disease severity across nine ASE-standard categories. We observed long-tailed distributions for most diseases' severity labels.

## A.6 ADDITIONAL RESULTS AND DETAILS ON VISION TASKS

### A.6.1 VIEW CLASSIFICATION DATASET AND BENCHMARKING

Transthoracic echocardiography (TTE) is a cornerstone of cardiac imaging for diagnosis and longitudinal follow-up. Routine TTE studies comprise many standardized views, yielding large volumes of cine loops for clinicians to review. Automatic view classification can accelerate retrieval of target clips and pre-populate structured reports. We evaluate this task on an internal, multi-vendor TTE dataset that spans diverse transducers, spatial/temporal resolutions, image quality, imaging depths, color Doppler, and contrast-enhanced studies. The dataset covers 18 American Society of Echocardiography (ASE)–style views (standard and contrast variants). All images were annotated by certified echocardiographers.

Below are the class abbreviations used in our figures:

- **A2C**: Apical two-chamber view.
- **A3C**: Apical three-chamber view.
- **A4C**: Apical four-chamber view.
- **A5C**: Apical five-chamber view.
- **Cont:A2C**: Apical two-chamber view with contrast.
- **Cont:A3C**: Apical three-chamber view with contrast.
- **Cont:A4C**: Apical two-chamber view with contrast.
- **Cont:SAX**: Parasternal short-axis view with contrast.
- **PLAX:ID**: Parasternal long-axis view with increased depth.
- **PLAX:LV**: Parasternal long-axis left ventricle .

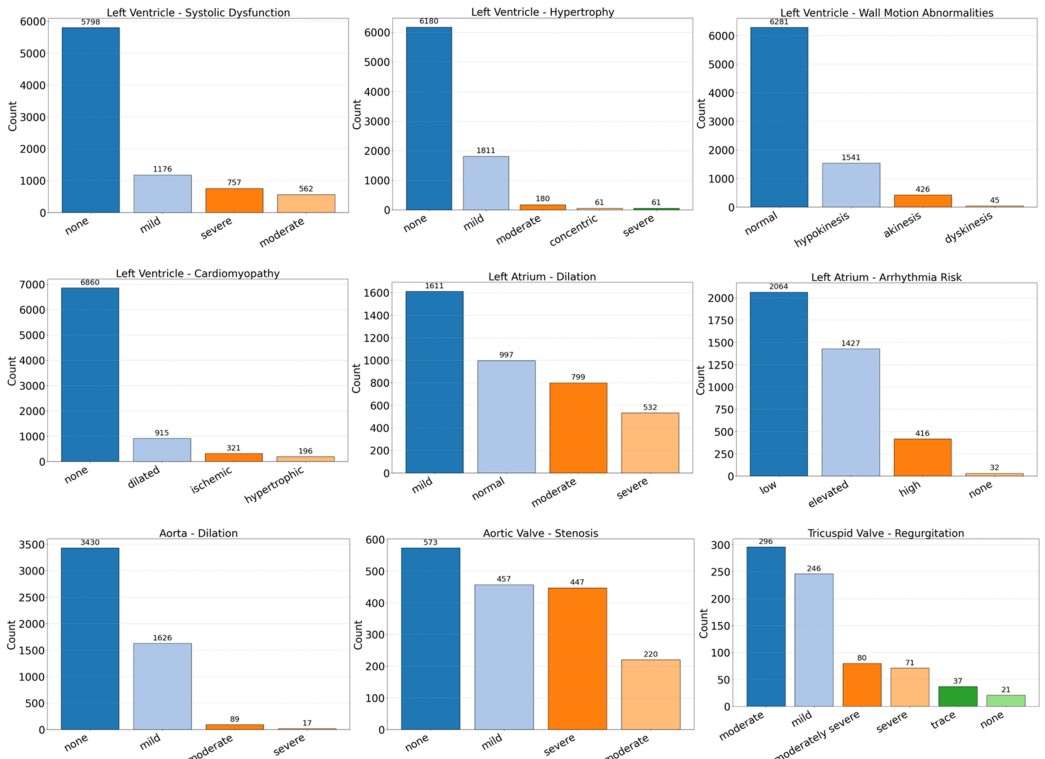

Figure 8: Fine-grained disease label distributions in EchoGround-MIMIC. For each of the 9 disease categories, we show the distribution of ASE-standard ordinal labels (e.g., none, mild, moderate, severe).

- **PLAX:RVN**: Parasternal long-axis right ventricle inflow.

- **PLAX:RVT**: Parasternal long-axis right ventricle outflow.

- **PLAX:VAL**: Parasternal long-axis zoomed on the mitral and/or aortic valve.

- **PSAX:AV**: Parasternal short-axis with a focus on the aortic valve.

- **PSAX:MV**: Parasternal short-axis with a focus on the mitral valve.

- **PSAX:PAP**: Parasternal short-axis at the papillary muscles level.

- **SC:4C**: Subcostal four-chamber view.

- **SC:IVC**: Subcostal long axis inferior vena cava view.

We next present quantitative and qualitative results for the view classification benchmark. Figure 10 shows the confusion matrix across the 18 echocardiographic views, where the strong diagonal pattern indicates high overall accuracy and residual errors are concentrated among anatomically adjacent views or contrast subtypes. Figure 11 reports class-wise balanced accuracy compared with baseline models, demonstrating that EchoVLM achieves superior performance over prior vision–language models while remaining competitive with vision-only foundation models.

A.7 GENERALIZATION ACROSS INSTITUTIONS AND IMAGING PROTOCOLS

Although EchoGround–MIMIC originates from a single institution, the downstream datasets used in our evaluation naturally span a broad range of acquisition settings, demographics, and ultrasound vendors. The public EchoNet datasets used for segmentation and landmark detection were collected in the United States, while the CAMUS dataset was acquired in France and reflects distinct imaging protocols and scanner characteristics.

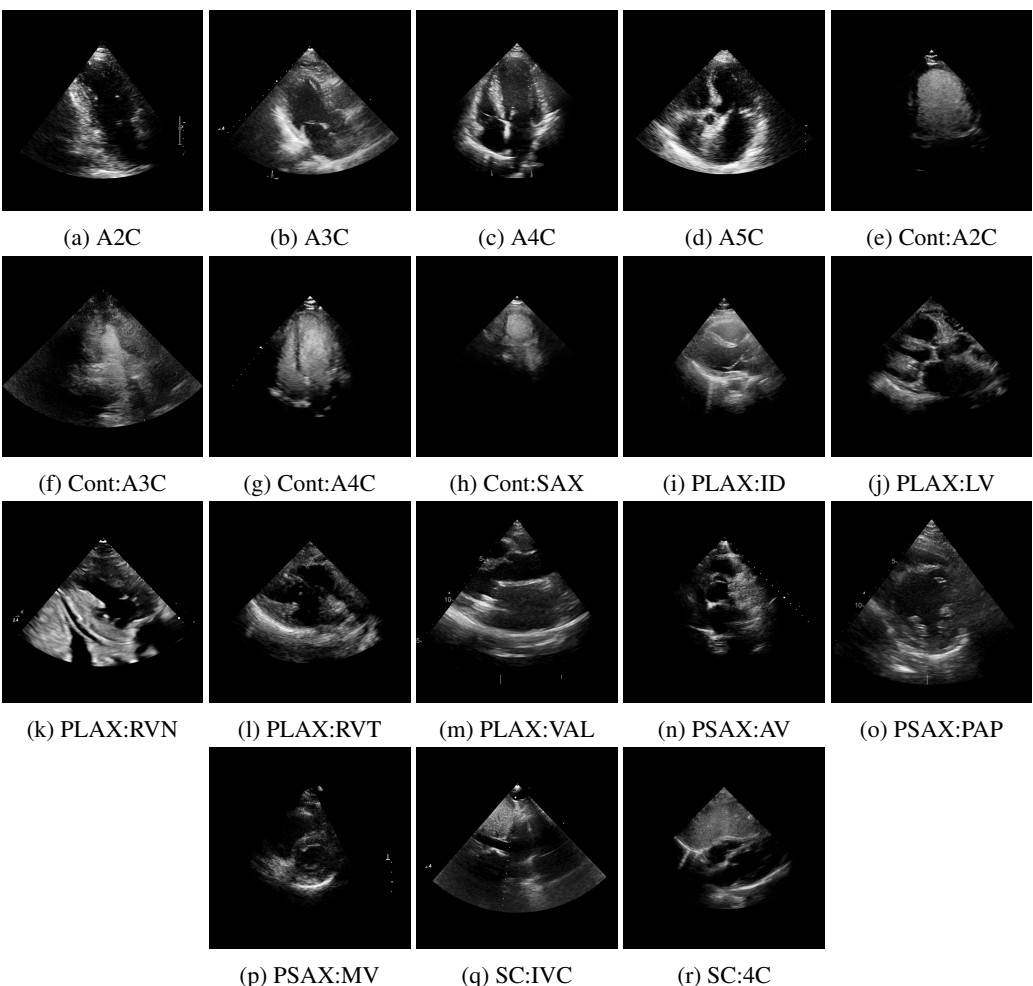

Figure 9: **Example of images used in the TTE view classification task.**

For view classification, our evaluation further incorporates studies from multiple clinical sites across different geographic regions and institutions. Table 9 summarizes performance across these sites, showing that EchoVLM maintains consistently strong accuracy, precision, and recall despite cross-institutional variability.

Table 9: Performance of EchoVLM across different clinical sites.

| Clinical Site | F1 | Precision | Recall |
|---|---|---|---|
| New York, US | 93.2 | 95.6 | 91.8 |
| Georgia, US | 96.3 | 96.4 | 96.7 |
| New Jersey, US | 95.9 | 97.1 | 95.2 |
| Minnesota, US | 91.8 | 92.2 | 92.5 |
| Asia | 91.1 | 95.1 | 89.9 |
| Europe (CAMUS, KNN) | 94.0 | 94.0 | 94.0 |

These results suggest that the visual representations learned by EchoVLM generalize well across heterogeneous imaging environments, even though pretraining was performed using a single-institution source dataset.

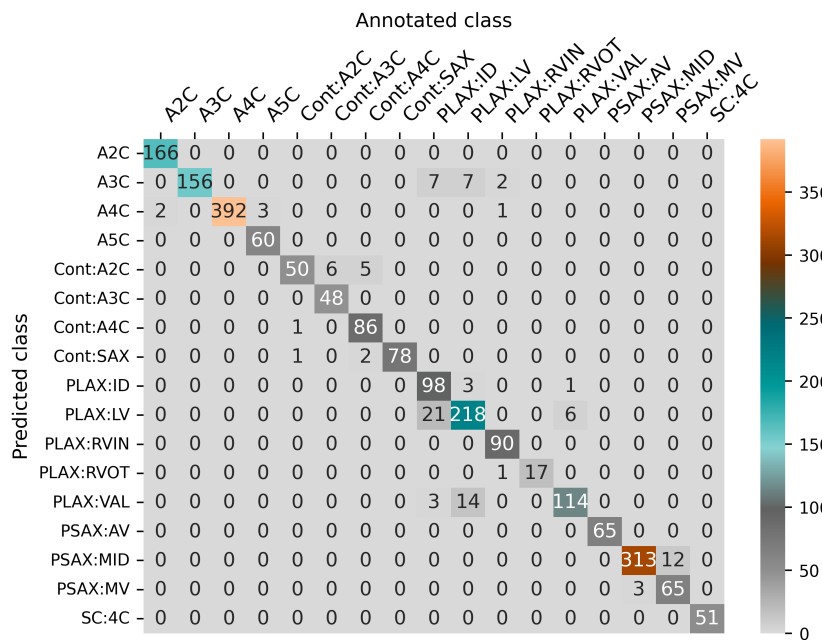

Figure 10: View classification confusion matrix (EchoVLM). Counts per class on the multi-vendor TTE test set (darker = higher). The strong diagonal indicates high accuracy; remaining errors are concentrated among anatomically adjacent views and contrast subtypes (e.g., PLAX and PSAX variants).

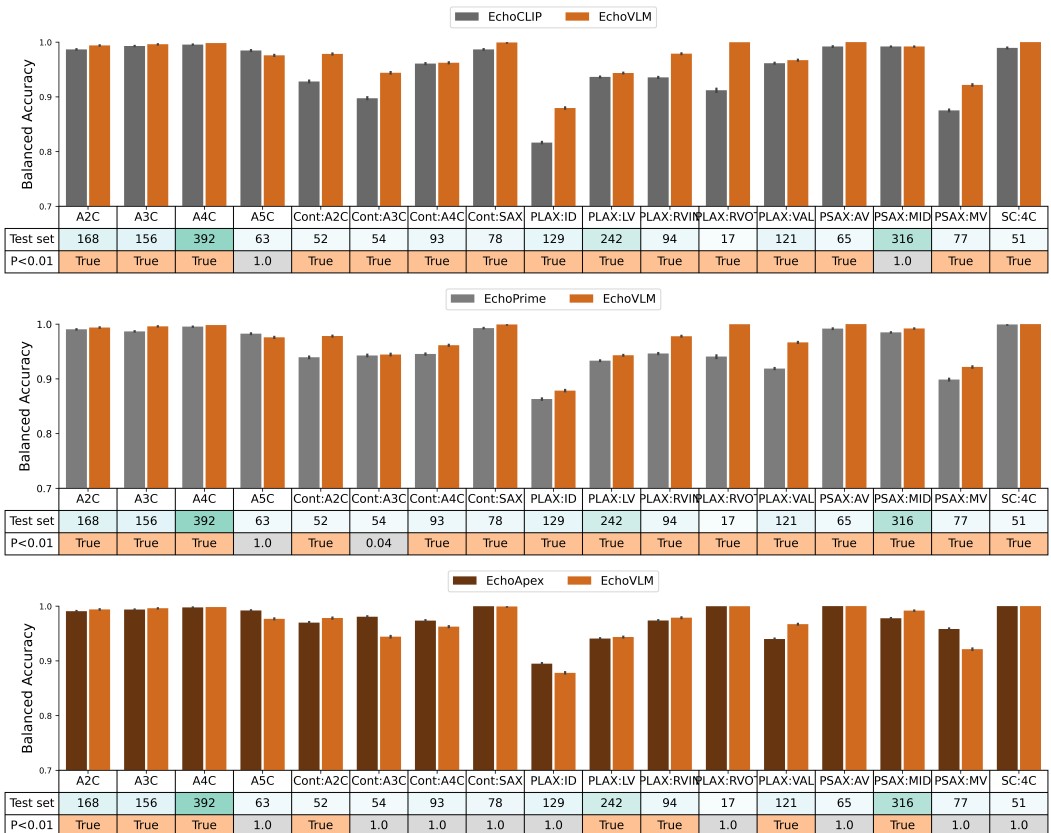

Figure 11: Class-wise view classification (balanced accuracy). Top: EchoVLM vs. EchoCLIP (VLM baseline). Middle: EchoVLM vs. EchoPrime. Bottom: EchoVLM vs. EchoApex (vision-only FM). Bars show balanced accuracy per ASE view on the multi-vendor TTE test set. Row "Test set" reports the number of test images per class; row "$p < 0.01$" marks whether EchoVLM's gain is significant (one-sided $t$-test). EchoVLM achieves higher balanced accuracy on most views than existing image-based VLM and video-based VLM while remaining competitive with vision-only FM.

## A.8 COMPARISON WITH THE VIDEO-BASED ECHOPRIME MODEL

**Fine-tuning comparison.** Because EchoPrime was originally designed as a video-level model, we evaluated two fine-tuning settings adapted to our frame-based benchmark: (1) *EchoPrime-FT-Sequence*, using 16 real consecutive frames; and (2) *EchoPrime-FT-Static*, repeating the same frame 16 times to match the single-frame information provided to EchoVLM. As shown below, EchoVLM achieves higher performance in all metrics.

| Model | F1 | Precision | Recall |
|---|---|---|---|
| EchoPrime-FT-Sequence | 93.8 | 94.9 | 93.2 |
| EchoPrime-FT-Static | 92.2 | 92.5 | 92.1 |
| EchoVLM (ours) | 95.3 | 95.8 | 95.1 |

**k-NN evaluation (no fine-tuning).** To isolate the quality of the learned visual representations and avoid fine-tuning–related confounders, we also performed a k-NN classification following the standard DINO evaluation protocol ($k = 20$, frozen backbone). We tested both our internal dataset and an unseen public dataset (CAMUS). EchoVLM again demonstrates stronger feature quality.

| Model | Dataset | F1 | Precision | Recall |
|---|---|---|---|---|
| EchoPrime | Internal-26k | 38.9 | 38.7 | 53.1 |
| EchoVLM (ours) | Internal-26k | 53.3 | 54.2 | 62.7 |
| EchoPrime | CAMUS-1k | 79.3 | 80.0 | 84.5 |
| EchoVLM (ours) | CAMUS-1k | 94.0 | 94.0 | 94.0 |

**Confusion matrices on CAMUS.** On the CAMUS test split (A2C/A4C videos), EchoVLM provides notably more accurate view discrimination, further indicating superior vision encoder representations.

| EchoPrime | A2C | A4C | | EchoVLM (ours) | A2C | A4C |
|---|---|---|---|---|---|---|
| A2C | 49 | 19 | | A2C | 47 | 3 |
| A4C | 1 | 31 | | A4C | 3 | 47 |

Across all evaluations—fine-tuning, k-NN representation quality, and cross-dataset generalization—the video-based EchoPrime model consistently underperforms the frame-based EchoVLM. These results suggest that EchoPrime is optimized for video–language aggregation rather than high-fidelity frame-level visual representation, whereas EchoVLM's design more effectively captures clinically relevant frame-based features.

| config | value |
|---|---|
| optimizer | AdamW |
| base learning rate | 5e-5 |
| weight decay | 0.005 |
| optimizer momentum | $\beta_1, \beta_2 = 0.9, 0.999$ |
| batch size | 512 |
| learning rate schedule | cosine decay |
| warmup epochs | 10 |
| training epochs | 100 |
| losses | CE |

Table 10: Hyperparameters used for the view classification experiment.

### A.8.1 STRUCTURE SEGMENTATION DATASET AND BENCHMARKING

We benchmark chamber segmentation on three public echocardiography datasets: EchoNet-Dynamic (Ouyang et al., 2020b), EchoNet-Pediatric (Zhang et al., 2021), and CAMUS (Leclerc et al., 2019a). These datasets span different populations and acquisition protocols, providing complementary challenges for evaluation. EchoNet-Dynamic contains over 10k apical four-chamber (A4C) videos with end-diastolic (ED) and end-systolic (ES) left ventricle masks. EchoNet-Pediatric comprises pediatric A4C and parasternal short-axis (PSAX) videos with left ventricle annotations. CAMUS provides 500 adult patients with A2C and A4C views annotated for both left ventricle and left atrium at ED/ES. Dataset statistics are summarized in Table 11.

We adapt EchoVLM for interactive segmentation by attaching a prompt-conditioned encoder–decoder following SAM (Kirillov et al., 2023). Models are trained with DiceCE loss and evaluated using the Dice similarity coefficient (DSC). As shown in Table 12, EchoVLM consistently outperforms task-specific baselines (U-Net, MedSAM) and performs on par with the vision foundation model EchoApex across all datasets. Notably, EchoVLM achieves the highest DSC on EchoNet-Dynamic (93.1%) and EchoNet-Pediatric A4C (92.4%), while matching EchoApex on EchoNet-Pediatric PSAX and CAMUS. These results indicate that measurement-grounded multi-modal pretraining preserves fine-grained local features required for precise structure segmentation, while also transferring across age groups and acquisition settings.

| Dataset | Structure | Views | Patients | Videos | Annotations | Eval. (ED, ES) |
|---|---|---|---|---|---|---|
| EchoNet-Dynamic | LV | A4C | 10,030 | 10,030 | 20,048 | 2,552 |
| EchoNet-Pediatric | LV | A4C | 1,958 | 3,176 | 6,449 | 1,386 |
| | LV | PSAX | – | 4,424 | 9,001 | 1,928 |
| CAMUS | LV | A4C | 500 | 500 | 9,964 | 80 |
| | LV | A2C | 500 | 500 | 9,268 | 80 |
| | LA | A4C | 500 | 500 | 9,964 | 80 |
| | LA | A2C | 500 | 500 | 9,264 | 80 |

Table 11: Details of datasets used for structure segmentation. We report the number of patients, videos, annotations, and evaluation cases at end-diastole (ED) and end-systole (ES).

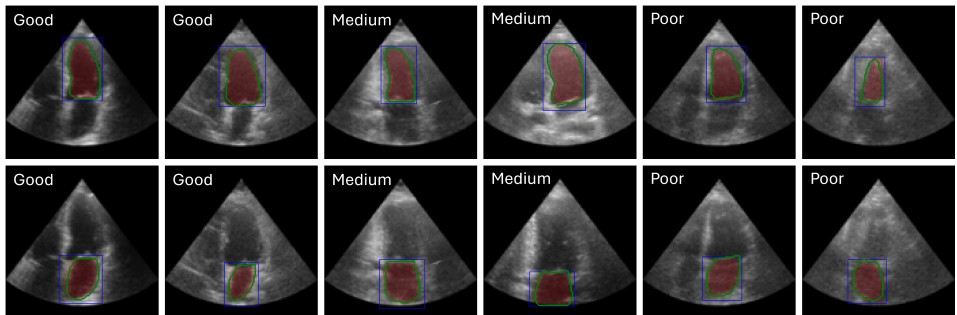

Figure 12: Additional EchoVLM segmentation results on CAMUS dataset. Image quality labels: Good, Medium, Poor. Image quality is assessed from the original dataset performed by CAMUS annotation team. Red = prediction, Green = annotation.

| Dataset | Target | Specialist | MedSAM | EchoApex | EchoVLM |
|---|---|---|---|---|---|
| EchoNet-Dynamic | LV-ES | 90.3 | 84.5 | 91.7 | **92.0** |
| | LV-ED | 92.7 | 88.5 | 93.9 | **94.1** |
| | ENDym LV mean | 91.5 | 86.5 | 92.8 | **93.1** |
| EchoNet-Pediatric | A4C-LV | 89.1 | 87.2 | 92.2 | **92.4** |
| | PSAX-LV | 89.6 | 88.0 | 93.0 | **93.1** |
| CAMUS | LV-ES | 91.6 | 85.6 | **93.0** | 92.9 |
| | LV-ED | 93.9 | 88.7 | 94.5 | **94.6** |
| | CAMUS LV mean | 92.8 | 87.2 | **93.8** | 93.8 |
| | LA-ES | 91.8 | 82.5 | **92.0** | 91.4 |
| | LA-ED | 88.9 | 78.0 | **89.6** | 89.0 |
| | CAMUS LA mean | 90.4 | 80.3 | **90.8** | 90.2 |

Table 12: Benchmark of segmentation performance (DSC) with state-of-the-art models. Bold indicates the best score in each row. Proposed EchoVLM model shows improvement over task-specialist models, medical generalist models and vision-only foundation models.

### A.8.2 LANDMARK DETECTION DATASET AND BENCHMARKING

We evaluate landmark detection on the public EchoNet-LVH dataset (Duffy et al., 2022), which benchmarks left ventricular hypertrophy (LVH) assessment from PLAX echocardiographic frames. The dataset provides frame-level annotations for three key structures: the interventricular septum (IVS), left ventricular internal dimension (LVID), and left ventricular posterior wall (LVPW). From these landmarks, wall thicknesses and cavity dimensions can be derived. Following the official split, we use 20,254 frames for training, 2,275 for validation, and 683 for testing.

| config | value |
|---|---|
| optimizer | AdamW |
| base learning rate | 5e-4 |
| weight decay | 0.005 |
| optimizer momentum | $\beta_1, \beta_2 = 0.9, 0.999$ |
| batch size | 64 |
| learning rate schedule | cosine decay |
| training epochs | 60 |
| losses | DiceCE |

Table 13: Hyperparameters used for the segmentation experiment.

We attach a UNETR-style decoder (Hatamizadeh et al., 2022) to EchoVLM for landmark prediction and train using a combination of GeneralizedDice and Focal losses (see Table 14). At inference, landmark coordinates are obtained from the predicted heatmaps as the centroids of thresholded activation regions. Performance is assessed using mean absolute error (MAE, in mm) of derived measurements and the average landmark error (Average L.E., Euclidean distance in mm) between predicted and ground-truth locations.

As summarized in Table 4, EchoVLM achieves competitive performance, outperforming both the task-specific baseline DeepLabV3 and the vision foundation model EchoApex on IVS and LVID measurements. These results confirm that our multimodal pretraining retains fine-grained spatial sensitivity required for clinical measurement tasks, providing reliable landmark localization in echocardiographic frames.

| config | value |
|---|---|
| optimizer | AdamW |
| base learning rate | 2e-4 |
| weight decay | 0.005 |
| optimizer momentum | $\beta_1, \beta_2 = 0.9, 0.999$ |
| batch size | 128 |
| learning rate schedule | cosine decay |
| training epochs | 150 |
| losses | GeneralizedDice + Focal |

Table 14: Hyperparameters used for landmark detection experiment.

## A.9 ADDITIONAL RESULTS AND DETAILS ON VISION-LANGUAGE TASKS

### A.9.1 IMPLEMENTATION DETAILS

We provide the full set of hyperparameters used for multimodal pretraining on EchoGround-MIMIC in Table 15. EchoVLM was trained with AdamW optimizer and a cosine decay learning rate schedule, with 200 warmup steps and a total of 20 epochs. The loss combined standard CLIP contrastive alignment with our proposed view-informed and negation-aware objectives, weighted by $\lambda_{\text{view}}$ and $\lambda_{\text{neg}}$. To ensure fair comparisons, we finetune each VLM on EchoGround-MIMIC following the same data split.

| config | value |
|---|---|
| optimizer | AdamW |
| base learning rate | 1e-4 |
| weight decay | 0.005 |
| optimizer momentum | $\beta_1, \beta_2 = 0.9, 0.999$ |
| batch size | 512 |
| learning rate schedule | cosine decay |
| warmup steps | 200 |
| training epochs | 20 |
| losses | $\mathcal{L}_{\text{CLIP}} + \lambda_{\text{view}}\mathcal{L}_{\text{view}} + \lambda_{\text{neg}}\mathcal{L}_{\text{neg}}$ |

Table 15: Hyperparameters used for pretraining on EchoGround-MIMIC.

### A.9.2 DISEASE ZEROSHOT RESULTS

We further report detailed results for disease zero-shot classification on the EchoGround-MIMIC test set. Figure 13 shows confusion matrices across the nine disease categories, highlighting that EchoVLM yields both high sensitivity for positive findings and improved discrimination of negative cases. Figure 14 presents additional evaluation metrics, confirming that EchoVLM consistently outperforms domain-specific and generalist VLM baselines across precision, recall, and AUC. These results demonstrate the effectiveness of grounding captions in quantitative measurements and enforcing negation-awareness, both of which are critical for accurate disease detection in clinical settings.

Figure 13: Confusion matrices for EchoVLM on EchoGround-MIMIC test set.

### A.9.3 VISUALIZATION OF ATTENTION MAPS

To better understand how EchoVLM attends to echocardiographic structures, we visualize CLS-token-to-patch attention maps from the final transformer block (Figure 15). Compared to CLIP, BiomedCLIP, and SigLIP2, EchoVLM produces sharper and more clinically meaningful attention patterns, focusing on cardiac chambers, septal boundaries, and valve regions relevant to the captions. This suggests that the proposed pretraining objectives not only improve task-level performance but also enhance interpretability of multimodal representations.

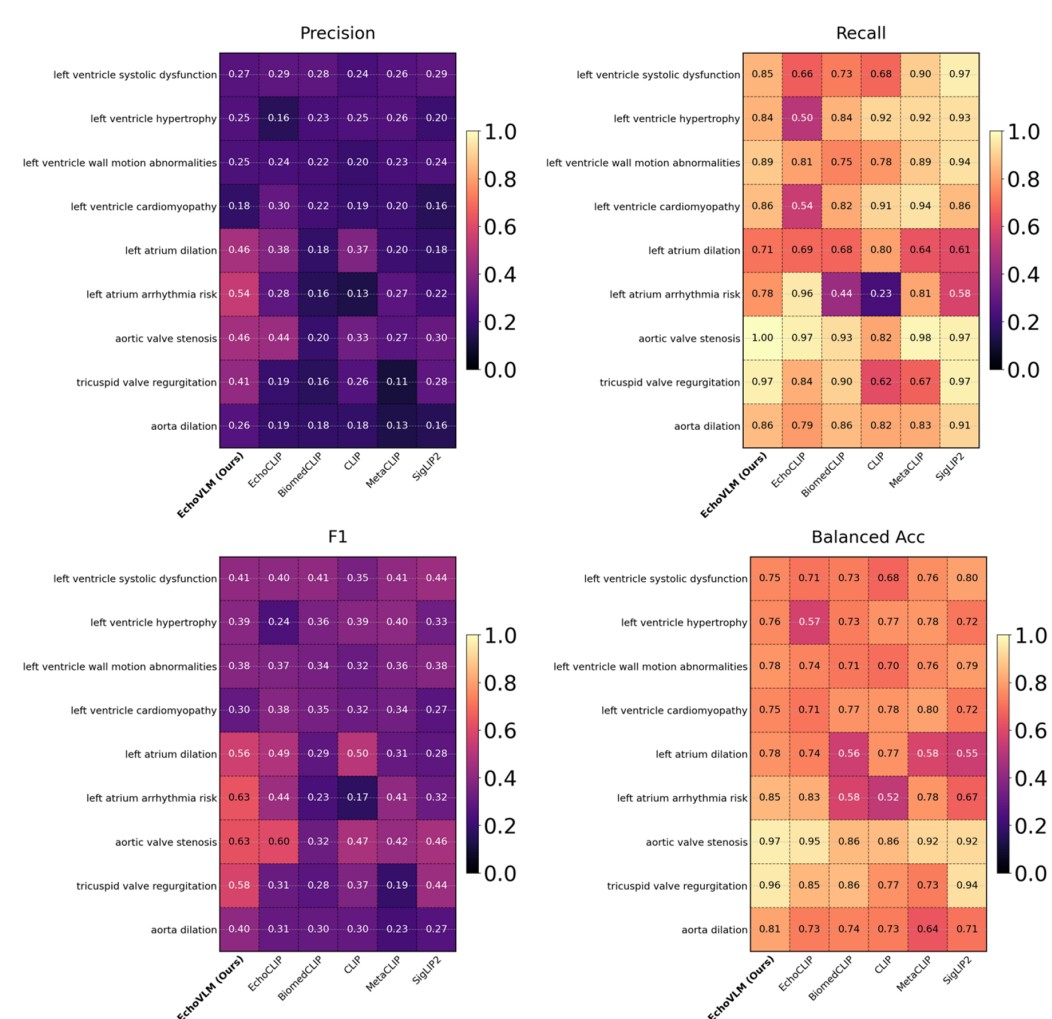

Figure 14: Additional results for disease zeroshot classification results on EchoGround-MIMIC test set.

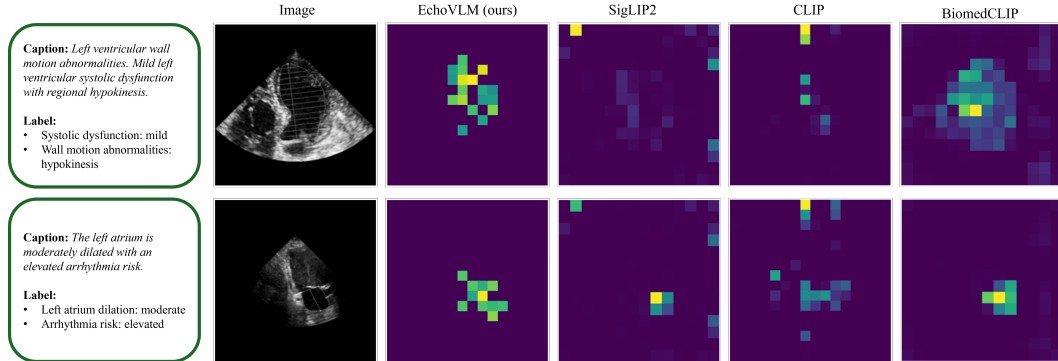

Figure 15: Qualitative comparison of CLS-token-to-patch attention maps on echocardiogram images. We extract self-attention weights from the final transformer block, select all rows corresponding to CLS-to-patch attention across heads, average them, and reshape to form attention maps. From left to right: caption with labels, input image, EchoVLM, EchoVLM (ours), SigLIP2, CLIP, BiomedCLIP.

### A.10 PSEUDO CODE FOR TRAINING

---

**Algorithm 1** EchoVLM Pretraining

---

**Require:** $\{(x_i, t_i, n_i, v_i)\}_{i=1}^{B}$ Batch of $B$ training samples, where
1: $x_i$: echocardiogram image,
2: $t_i$: measurement-grounded caption,
3: $n_i$: negated caption,
4: $v_i$: view label.
5: $f_I, f_T$: image and text encoders.
6: $\tau$: temperature.
7: $\lambda_{\text{view}}, \lambda_{\text{neg}}$ Loss weights for view-informed and negation-aware objectives.
8: Opt optimizer.

9: **function** COMPUTE_ECHOVLM_STEP($f_I, f_T, \tau, \lambda_{\text{view}}, \lambda_{\text{neg}}, \text{Opt}$)
10: $\quad z_i^I \leftarrow f_I(x_i), \ z_i^T \leftarrow f_T(t_i), \ z_i^N \leftarrow f_T(n_i)$ $\qquad\qquad\qquad$ ▷ Encode image and captions
11: $\quad Z_I \leftarrow [z_1^I; \ldots; z_B^I], \ Z_T \leftarrow [z_1^T; \ldots; z_B^T], \ Z_N \leftarrow [z_1^N; \ldots; z_B^N]$
12: $\quad v \leftarrow [v_1; \ldots; v_B]$ $\qquad\qquad\qquad\qquad\qquad\qquad\qquad\qquad$ ▷ Batch view-label
13: $\quad S_{IT} \leftarrow \tau Z_I Z_T^\top, \quad S_{TI} \leftarrow S_{IT}^\top$ $\qquad\qquad\qquad\qquad$ ▷ Image–text similarities
14: $\quad S_{II} \leftarrow \tau Z_I Z_I^\top; \ \text{set} \ (S_{II})_{ii} \leftarrow -\infty$ $\qquad$ ▷ Image–image similarities (no self similarity)
15: $\quad L_{\text{CLIP}} \leftarrow \text{CLIP\_LOSS}(S_{IT}, S_{TI})$ $\qquad\qquad\qquad\qquad$ ▷ Symmetric image↔text CE
16: $\quad L_{\text{view}} \leftarrow \text{VIEW\_CONTRASTIVE}(S_{II}, v)$ $\qquad\qquad\qquad$ ▷ View-contrastive loss
17: $\quad L_{\text{neg}} \leftarrow \text{NEG\_BCE}(Z_T, Z_N, \tau)$ $\qquad\qquad\qquad$ ▷ Separate pos vs neg caption
18: $\quad L \leftarrow L_{\text{CLIP}} + \lambda_{\text{view}} L_{\text{view}} + \lambda_{\text{neg}} L_{\text{neg}}$ $\qquad\qquad\qquad\qquad$ ▷ Final objective
19: $\quad$ **Backward** $L$ and update $f_I, f_T$ with Opt $\qquad\qquad\qquad$ ▷ Update the network
20: $\quad$ **return** $L$
21: **end function**

22: **function** CLIP_LOSS($S_{IT}, S_{TI}$)
23: $\quad L_1 \leftarrow \text{CE\_ROW}(S_{IT})$ $\qquad\qquad\qquad\qquad\qquad$ ▷ row-wise CE w.r.t. diagonal
24: $\quad L_2 \leftarrow \text{CE\_ROW}(S_{TI})$ $\qquad\qquad\qquad\qquad\qquad$ ▷ row-wise CE w.r.t. diagonal
25: $\quad$ **return** $\frac{1}{2}(L_1 + L_2)$
26: **end function**

27: **function** VIEW_CONTRASTIVE($S_{II}, v$) $\qquad\qquad\qquad\qquad$ ▷ contrastive loss over views
28: $\quad M_{ij} \leftarrow \mathbb{1}[v_i = v_j \ \wedge \ i \neq j]$ $\qquad\qquad\qquad$ ▷ mask: same-view positives
29: $\quad$ **for** $i = 1$ to $B$ **do**
30: $\quad\quad d_i \leftarrow \max(1, \sum_j M_{ij})$ $\qquad\qquad\qquad\qquad$ ▷ number of positives for $i$
31: $\quad\quad \ell_i \leftarrow -\frac{1}{d_i} \sum_j M_{ij} \log \frac{\exp((S_{II})_{ij})}{\sum_{k \neq i} \exp((S_{II})_{ik})}$ $\qquad\qquad$ ▷ per-sample loss
32: $\quad$ **end for**
33: $\quad$ **return** $\frac{1}{B} \sum_{i=1}^{B} \ell_i$ $\qquad\qquad\qquad\qquad\qquad$ ▷ average over batch
34: **end function**

35: **function** NEG_BCE($Z_T, Z_N, \tau$) $\qquad\qquad\qquad\qquad\qquad$ ▷ negation-aware loss
36: $\quad u_i \leftarrow \tau \langle z_i^T, z_i^N \rangle$ $\qquad\qquad\qquad$ ▷ logit between caption and its negation
37: $\quad$ **return** $\frac{1}{B} \sum_{i=1}^{B} \text{BCEWithLogits}(u_i, 0)$ $\qquad\qquad$ ▷ push apart each $(t_i, n_i)$ pair
38: **end function**

39: **function** CE_ROW($S$) $\qquad\qquad\qquad\qquad\qquad\qquad$ ▷ row-wise CE to diagonal
40: $\quad$ **return** $\frac{1}{B} \sum_{i=1}^{B} \left( -\log \frac{\exp(S_{ii})}{\sum_{j=1}^{B} \exp(S_{ij})} \right)$
41: **end function**

---