# OpenReview forum: "EchoVLM: Measurement-Grounded Multimodal Learning for Echocardiography"
_ICLR.cc/2026/Conference — ICLR 2026 Conference Withdrawn Submission_

### Official Review · Reviewer_qUxR · 2025-10-17

**Soundness:** 2
**Presentation:** 3
**Contribution:** 3
**Rating:** 4
**Confidence:** 3

**Summary:**

The paper introduces EchoVLM, a measurement-grounded VLM for echocardiography, and EchoGround-MIMIC, a multimodal dataset explicitly linking echocardiographic images with structured measurements, measurement-grounded captions, and guideline-aligned disease labels. EchoVLM extends the CLIP framework with two clinically motivated objectives: a view-informed contrastive loss that models the view-dependent nature of echocardiographic images, and a negation-aware contrastive loss that distinguishes positive and negative clinical statements. Trained on 19,065 image–text pairs, EchoVLM achieves state-of-the-art results across 36 tasks, including zero-shot disease classification (AUC = 86.5%), view classification (95.1% accuracy), and competitive segmentation and landmark detection on public datasets. The results demonstrate strong cross-modal transfer and clinically meaningful visual representations.

**Strengths:**

- New measurement-grounded dataset: EchoGround-MIMIC provides the first large-scale, structured dataset pairing echo images with quantitative measurements, standardized views, and guideline-derived labels.

- The proposed view-informed and negation-aware contrastive losses directly encode clinical reasoning patterns, improving both visual coherence and semantic discrimination.

- Extensive validation on 36 tasks across five applications shows consistent superiority over domain and generalist baselines (e.g., +7.2 AUC vs. EchoCLIP, +0.9% precision over EchoApex).

**Weaknesses:**

- EchoGround-MIMIC originates from a single institution (MIMIC-IV-ECHO), which constrains demographic, hardware, and acquisition variability, potentially limiting generalization.

- OCR-extracted measurements and LLM-generated captions introduce potential noise; limited manual validation may not fully prevent systematic errors.

- While effective, the new contrastive objectives are empirically motivated with limited theoretical analysis of their convergence or interaction with the CLIP loss.

- EchoVLM operates on single frames rather than sequences, missing dynamic cardiac context critical for echocardiographic interpretation.

**Questions:**

- How would EchoVLM perform when trained or tested on multi-institutional data with varying imaging protocols, demographics, and ultrasound vendors?

- Could the framework be extended to incorporate video-level dynamics, given that echocardiography interpretation heavily depends on temporal motion?

- What proportion of the automatically generated measurement-grounded captions were manually verified, and how sensitive are downstream results to errors in this supervision?

- How robust are the view-informed and negation-aware losses to their respective λ parameters when scaling to larger datasets or different imaging domains?

- What are the major failure cases observed in zero-shot classification, e.g., misinterpretations of negations or confusion between anatomically adjacent views?

- What are the computational requirements and latency of EchoVLM inference in a real-time clinical setting, and how might model compression or distillation affect its performance?

---

> ### Author Response · Authors · 2025-11-26
> **Response to Reviewer qUxR Part 1/4**
>
> > W1, Q1. EchoGround-MIMIC originates from a single institution... Generalization to multi-institutional data
>
> We thank the reviewer for raising this important point regarding generalizability across institutions, demographics, and imaging protocols. While EchoGround-MIMIC originates from a single institution, **the downstream datasets used in our evaluation naturally incorporate substantial variability**. For example, the public EchoNet datasets used for segmentation and landmark detection were collected in the United States, whereas the CAMUS dataset was collected in France (Europe). In addition, the view-classification data used in our experiments includes studies sourced from multiple clinical sites across different geographic regions and ultrasound vendors:
>
> | Clinical Site       | F1   | Precision | Recall |
> |---------------------|------|-----------|--------|
> | New York, US        | 93.2 | 95.6      | 91.8   |
> | Georgia, US         | 96.3 | 96.4      | 96.7   |
> | New Jersey, US      | 95.9 | 97.1      | 95.2   |
> | Minnesota, US       | 91.8 | 92.2      | 92.5   |
> | Asia                | 91.1 | 95.1      | 89.9   |
> | Europe (CAMUS, knn) | 94.0 | 94.0      | 94.0   |
>
> The CAMUS row corresponds to a zero-finetuning evaluation that measures the generalization ability of our pretrained vision backbone. Here we freeze the encoder and apply a k-nearest-neighbor classifier directly on the (unseen) CAMUS dataset. EchoVLM achieves strong performance (F1 = 94.0), demonstrating that the learned representations transfer well to a different continent, different vendor systems, and different acquisition protocols.
>
> Across all sites—including multi-institutional U.S. data, an Asia-based cohort, and the European CAMUS dataset—EchoVLM maintains consistently high performance.
>
> > W3. Limited theoretical analysis.
>
> We appreciate the reviewer’s observation. Our proposed objectives are grounded in echo-specific inductive biases (i.e. view structure and clinical negation) which are not captured in standard theoretical CLIP formulations. We provide the following intuitions and analysis for our added objectives:
>
> Our intuition is standard CLIP loss cannot handle cases where more than one sample is known to belong to the same underlying semantic class. The default CLIP objective will still push the intra-class positives apart, leading to false negatives. This is suboptimal in echocardiography, where the data are highly view-centric: many frames from the same study (and across studies) correspond to the same ASE view (e.g., A4C, PLAX) and should form tight clusters in the embedding space. Therefore, a straightforward approach to encode all images sharing the the same view as positives and all others as negatives. This gives rise to our view-informed contrastive loss:
>
> $$
> L\_{\mathrm{view}} = -\\frac{1}{N} \\sum\_{i=1}^{N} \\frac{1}{|P(i)|} \\sum\_{j \\in P(i)}
> \\log \\frac{\\exp(\\tau \\mathbf{z}\_{i} \\cdot \\mathbf{z}\_{j})}
> {\\sum\_{k \\neq i} \\exp(\\tau \\mathbf{z}\_{i} \\cdot \\mathbf{z}\_{k})}
> $$
>
>
> where $\mathbf{z}_{i}$  is the image embedding, P(i) is the set of positives sharing the same view as i, and the sum in the denominator runs over all other images in the batch (positives + negatives).
>
> Regarding its connection to CLIP, we claim that CLIP is a special case of view-contrastive loss where the labels are instance IDs (i.e. each image-text pair is its own class). Consider a batch of $N$ image–text pairs $(x_i, t_i)$. Let
> $$
> z_i^{I} = f_I(x_i), \qquad z_i^{T} = f_T(t_i)
> $$
> be the image and text embeddings. If we let label be the each index \(i\) and the two views become the image embedding $z_i^{I}$ and text embedding $z_i^{T}$, then using the same formulation for view-contrastive loss yields:
>
> $$
> L_{I\to T} = -\frac{1}{N} \sum_{i=1}^N \frac{1}{|P(i)|}
> \log \frac{\exp(\tau\, z_i^{I} \cdot z_i^{T})}
> {{\sum_{j=1}^N} \exp(\tau\, z_i^{I} \cdot z_j^{T})}.
> $$
> which is the standard CLIP image-to-text loss. From this perspective, a near-optimal CLIP solution is one where each image–text pair forms a tight cluster and different pairs are spread out in the joint embedding space. However, this does not impose any constraint that images from the same ASE view (but different patients) should be close in the vision embedding space, as the text embeddings for same-view patients may be quite different. Regarding convergence, since both objectives are smooth and bounded (greater than or equal to 0), combining them still results in a smooth and non-convex objective. Under the usual assumptions for stochastic gradient descent used in deep learning training, this guarantees convergence to a first-order stationary point of the joint objective.

---

> ### Author Response · Authors · 2025-11-26
> **Response to Reviewer qUxR Part 2/4**
>
> > W2. Q3. OCR-extracted measurements and LLM-generated captions introduce potential noise... What proportion of the automatically generated measurement-grounded captions were manually verified ...
>
> **OCR Measurement Extraction and Cleaning.** The OCR step is used exclusively to extract measurement name–value pairs. After the first pass, we found `1,232 unique keys`—many of which represented the same measurement with slightly different naming conventions (e.g., “AV_Vmax” vs. “AV Vmax”), while others corresponded to non-clinical overlays such as “gain,” “velocity scale,” or “time scale.” We focused on key–value pairs appearing more than 10 times, resulting in `278 candidates`. Based on clinical practice, all measurements should fall under `11 categories` (LV, LA, RV, RA, MV, TV, AV, PV, SV, PVein, Aorta). Our team manually reviewed these categories and cleaned/classified the 278 candidates into `167 final structured measurements`, which are released in our repository organized in pydantic classes (see `data/mimic_anatomy.py` in shared repo).
>
> This manual cleanup significantly improved downstream reliability. After consolidation, we observed that the LLM almost never mis-parsed the cleaned measurement strings into structured dictionaries. The final dataset includes structured metadata for each measurement (anatomy, name, value, unit, cardiac phase). We are also releasing the intermediate cleaning results for full transparency.
>
> **Verification of LLM-Generated Captions.** We collaborated closely with a cardiologist to guide the measurement curation and establish normal ranges for relevant quantitative measurements. To validate the LLM-generated disease captions, we implemented a rule-based consistency check: for each disease, and for each measurement associated with that disease, we compared the LLM’s disease label against a rule-based label derived from the numerical value.
>
> We found strong alignment in the majority of cases (approximately `87%`), where values were clearly within normal or abnormal ranges. The remaining `~13%` of cases involved borderline values or clinically subjective thresholds. For example, although the normal LA length is typically 3.5–5.2 cm, an LA length of 4.9 cm may still be described as “dilated” in the clinical report due to patient-specific factors such as age, comorbidities, or longitudinal trends unavailable in isolated measurements. In such cases, we retain the report-derived label but mark these samples with a binary "matched_with_LLM" flag in our JSON dataset to allow users to re-filter or re-interpret them as needed.
>
> During manual verification, we also removed diseases related to the mitral valve, mitral regurgitation, and stroke volume where measurement–caption inconsistencies were most prominent. They also rely on multi-parameter Doppler criteria and hemodynamic context that are not available in our context. To ensure dataset quality and reduce clinically implausible labels, we exclude these categories from the measurement-grounded disease labels.
>
> **Sensitivity in downstream tasks**. We have implemented a baseline analysis in which we directly use the uncurated clinical report as the language supervision for training EchoVLM. We did not use additional supervision strategies such as view contrastive or text negation loss. We used the same training and evaluation split and pretraining setup as in our main experiment. As seen below, training on raw reports leads to substantially worse multimodal alignment. In zero-shot disease classification, AUC drops from `79.6` to `54.4`, and precision drops from `29.00` to `12.47`. We suspect the reason for such decline is that raw clinical reports contain large amounts of non-specific information that impedes image-text alignment. Therefore, we leverage OCR-extracted measurements and ASE guideline to curate both visually and clinically-grounded captions. We believe that our curated EchoGround-MIMIC provides substantial and necessary benefit for training a clinically reliable VLM.
>
> | Data Source       | AUC    | Precision | Recall  | Recall@5 | Recall@10 |
> |-------------------|--------|-----------|---------|----------|-----------|
> | Raw Reports       | 54.43 | 12.47    | 66.62  | 0.31   | 0.78    |
> | Curated Captions  | 79.60 | 29.00    | 73.60  | 2.30   | 4.33    |

---

> ### Author Response · Authors · 2025-11-26
> **Response to Reviewer qUxR Part 3/4**
>
> > W4, Q2. EchoVLM operates on single frames rather than sequences ... Could the framework be extended to incorporate video-level dynamics ...?
>
> We thank you for raising this important question. There are several practical and empirical reasons why we adopt a frame-based approach for EchoVLM.
>
> 1. **Broader and more reliable data coverage.** Real-world echocardiography exams (including MIMIC-IV) are highly heterogeneous—many studies contain variable-length clips, missing frames, or only saved keyframes (e.g., ED/ES). A frame-based model can utilize all such data, whereas video models typically require complete and well-formed sequences, which greatly limits usable volume. This is especially important in our setting because many quantitative measurements are performed on—and clinically defined by—a single representative frame rather than a full sequence.
>
> 2. **Alignment with real clinical workflows.** Cardiologists frequently base diagnostic decisions on representative frames rather than full cine loops (e.g., evaluating chamber size from a static view). A frame-level representation aligns directly with how quantitative measurements and diagnostic statements are generated in practice.
>
> 3. **Experimental evidence**. State-of-the-art video models in Echo domain do not outperform frame-based VLMs. We compared EchoVLM (frame-based) against **EchoPrime (video-based)** to understand whether a video architecture yields advantages for the downstream tasks studied in our paper. The results are presented as follows.
>
> 3.1 Disease Classification and Retrieval.
>
> We adapted EchoPrime to our image–text setting by converting each image into a pseudo-video (3×16×224×224) and evaluating it under the exact same data splits and protocols. We trained and evaluated EchoPrime on the same EchoGround-MIMIC splits and protocols used for all other VLMs.
>
> As shown below, EchoPrime performed worse than EchoVLM in both disease zero-shot classification and image–text retrieval: disease AUC = 82.8 (vs. 86.5 for EchoVLM), and retrieval recall-5 = 1.4% (vs. 3.0% for EchoVLM). These results confirm that EchoPrime is not optimized for image–text alignment or retrieval tasks, consistent with its design as a video-level report aggregation model rather than a CLIP-style VLM.
>
> We note that training EchoPrime incurred **16 times higher memory requirements** compared with our image-based EchoVLM, so we did not use it as the main comparison baseline.
>
>
> | Model      | AUC      | Precision | Recall     | Recall@5    | Recall@10   |
> |------------|----------|-----------|------------|-------------|-------------|
> | EchoPrime  | 82.8 | 29.7  | 93.8   | 1.41    | 2.78    |
> | EchoVLM (ours)    | 86.5 | 34.2  | 86.2   | 2.98    | 5.70    |
>
>
> 3.2 Vision Task Evaluation (Fine-tuning)
>
> We further evaluated EchoPrime’s vision encoder on the downstream task of view classification. We considered two settings: 1) EchoPrime-FT-Sequence: fine-tuned with 16 real consecutive frames; 2) EchoPrime-FT-Static: fine-tuned with the same frame repeated 16× (to match same information received by image-based models). The results are as follows. Even with sequential frames, EchoPrime remains below EchoVLM.
>
>
> |                       |  F1  | Precision | Recall |
> |:---------------------:|:----:|:---------:|--------|
> | EchoPrime-FT-Sequence | 93.8 |    94.9   | 93.2   |
> | EchoPrime-FT-Static   | 92.2 |    92.5   | 92.1   |
> | EchoVLM (ours)        | 95.3 |    95.8   | 95.1   |
>
>
> From these analysis, our observation is that currently video-based model does not outperform frame-based approach. Meanwhile, we also emphasize that **our choice of a frame-based model in this work should not be interpreted as our final solution** for echocardiography applications. Echocardiography is inherently dynamic, and many clinical tasks benefit from temporal information. Because of the heterogeneous and incomplete nature of real-world echo studies, we believe that a **hybrid** approach—combining both frame-based and video-based representations—may ultimately be the most powerful and clinically aligned strategy. Designing such a hybrid system requires **careful architectural considerations** to balance performance, robustness, and computational efficiency. This is an exciting direction, and we view it as important future work building upon EchoVLM.

---

> > ### Author Response · Authors · 2025-11-26
> > **Response to Reviewer qUxR Part 4/4**
> >
> > > Q4. How robust are the view-informed and negation-aware losses to their respective $\lambda$ parameters
> >
> > We thank you for raising the question. Our ablations in Table 6 demonstrate that the proposed objectives remain stable across a range of lambda values, and that performance varies smoothly, as shown below,
> >
> > | $\lambda_{\text{view}}$ | $\lambda_{\text{neg}}$ |  AUC | Precision | Recall |
> > |:-----------------------:|:----------------------:|:----:|:---------:|:------:|
> > |           1.0           |           1.0          | 79.6 |    29.0   |  73.6  |
> > |           0.25          |           0.5          | 81.8 |    28.0   |  87.2  |
> > |           0.5           |           0.5          | 78.1 |    26.3   |  81.3  |
> > |           0.5           |           0.1          | 83.9 |    32.2   |  83.4  |
> >
> > Regarding dataset scale, EchoGround-MIMIC is **among one of the largest open-source echocardiography image–text dataset** (`19,065` curated pairs with structured measurements and guideline-aligned labels). All of our ablation experiments are conducted at this scale. Regarding transfer to new imaging domains, the view-informed loss operates on ASE-standard views and are universally used across echo protocols. The negation-aware loss is based on language-level counterfactuals (e.g., “no regurgitation” vs. “mild regurgitation”), which are domain-agnostic. Therefore, we believe our pretraining objectives should transfer well to new echo imaging distributions (e.g., multi-center or vendor-specific data). **From our analysis on multi-institution evaluation, we did not observe significant performance variations**. These results indicate that our pretraining objective is not overly sensitive to dataset scales and demographics, supporting its practical robustness for real-world multimodal echo pretraining.
> >
> > > Q5 major failure cases in zero-shot classification ...
> >
> > We thank you for the insightful question. We analyzed EchoVLM’s zero-shot disease classification and identified key failure modes, particularly false positives in left ventricle (LV) diseases. Although all four LV disease classes achieve strong AUC (79–82%) and recall (84–89%), precision remains lower (18–27%). Cardiomyopathy is the most challenging class, partly due to the limited number of positive examples. Many false positives occur in studies showing LV abnormalities (e.g., systolic dysfunction or hypertrophy) but without cardiomyopathy. For instance, one FP study might state “mild symmetric left ventricular hypertrophy with normal cavity size and preserved systolic function”.
> >
> > This reflects a core challenge in echocardiography: **LV phenotypes often co-occur and exhibit overlapping presentations, making boundary distinctions difficult and leading to anatomically clustered errors**. Despite this, EchoVLM delivers meaningful improvements in AUC (+7.2%) and view classification accuracy (95.1%) over strong baselines. We view these failure modes as opportunities for refinement rather than fundamental limitations.
> >
> > > Q6 Computational requirements and latency of EchoVLM inference ... how might model compression or distillation affect its performance
> >
> > We thank the reviewer for question on inference efficiency in clinical workflows. We conducted a profiling study on an NVIDIA A100 GPU to quantify EchoVLM’s end-to-end latency and memory footprint across all downstream tasks. As summarized below, EchoVLM supports fast inference speed for all applications as we use light weighted decoder designs for most of the downstream tasks. The required peak GPU memory also is generally below 5GB.
> >
> > | Task                   | Model Size | Decoder Type | Inferencing Speed | Inferencing Memory |
> > |------------------------|------------|--------------|--------------------------|---------------------------|
> > | Disease Classification | 393.9M     | CLIP         | 251ms                    | 4.19GB                    |
> > | View Classification    | 86.6M      | Linear       | 5 ms                     | 0.83GB                    |
> > | Segmentation           | 105.8M     | SAM          | 9.5ms                    | 2.15GB                    |
> > | Landmark Detection     | 99.4M      | UNetr        | 15.3 ms                  | 2.49GB                    |
> >
> > We also emphasize that EchoVLM is designed primarily to augment the **diagnostic workflow** (instead of surgical interventions), which typically **requires tens of minutes to over an hour for a cardiologist to complete**. In this context, a `~250 ms` inference time is negligible relative to human workflow latency. Therefore, EchoVLM’s computational footprint is fully compatible with real-world clinical usage, both for **image interpretation** and for **post-exam review**. Additional model compression or distillation would primarily be needed for real-time interventional guidance, which is beyond the scope of our current target applications.

---

### Official Review · Reviewer_v1CK · 2025-10-29

**Soundness:** 2
**Presentation:** 3
**Contribution:** 3
**Rating:** 4
**Confidence:** 3

**Summary:**

This paper introduces EchoGround-MIMIC, a measurement-grounded multimodal echocardiography dataset, which includes standardized views, structured measurements, measurement-grounded captions, and guideline-derived disease labels. The authors also propose EchoVLM, a CLIP-style vision-language model, which is pretrained with two novel contrastive loss functions (view-informed and negation-aware).

**Strengths:**

1. A significant contribution of this work is the design of a comprehensive data processing pipeline. This pipeline successfully extracts and aligns a complex, multimodal dataset—comprising images, standardized views, quantitative measurements, measurement-related reports, and disease labels—from the MIMIC-IV-ECHO and MIMIC-IV-Note databases.

2. The paper proposes two novel and clinically-motivated pretraining objectives: view-informed contrastive learning and negation-aware contrastive learning. The utility and effectiveness of these objectives are well-supported by the provided ablation studies.

3. The proposed model (EchoVLM) is thoroughly evaluated on a diverse set of five downstream application types (36 tasks in total), demonstrating its generalizability and strong performance across both multimodal and vision-only benchmarks.

**Weaknesses:**

1. Disconnect between "Measurement-Grounded" Narrative and Methodology: The paper's core theme is "measurement-grounded multimodal learning." However, there appears to be a significant disconnect between this narrative and the technical implementation. The structured measurements (e.g., JSON-formatted values like "EF: 45%"), which are a key highlight of the new dataset, are not directly utilized as an input during the model's training phase. The model is only trained on the captions derived from these measurements. Furthermore, the two novel optimization objectives (L_view and L_neg)  are independent of the structured measurements. In fact, these objectives appear to be entirely separable from the 'grounded' nature of the data pipeline: L_view is a vision-only objective, while L_neg is a text-only objective that could be applied to any positive/negative caption pair, whether it is measurement-grounded or not.

2. Lack of Ablation on the "Grounded" Data Pipeline: While the "measurement-grounded" nature of the captions is presented as a core advantage, the paper lacks a direct ablation study to quantify the benefit of this complex and costly data curation process. Specifically, there is no experiment comparing the performance of a model trained on these curated "measurement-grounded captions" against a baseline model trained on the original, complete, and noisy "non-measurement-grounded" reports (e.g., the full text from MIMIC-IV-Note). This makes it difficult to assess the true value added by the grounding pipeline.

**Questions:**

1. Given the paper's core theme of "measurement-grounded" learning, could the authors elaborate on the design rationale for not directly utilizing the extracted structured measurements as an input during pretraining (e.g., as explicit tokens or an auxiliary regression loss)? Why was this quantitative data, a key part of the new dataset, only used as an intermediate tool for caption generation?

2. The paper provides a simple example for negation generation (e.g., "no regurgitation" from "mild regurgitation"). For more complex quantitative statements, such as "Quantitative biplane left ventricular ejection fraction is 45 %," could the authors clarify what form the corresponding "clinical semantic negation" takes?

3. Considering that echocardiography is an inherently dynamic (video-based) modality, what are the specific advantages or benefits of the proposed frame-based approach when compared to existing video-based solutions (such as EchoPrime mentioned in the related work)?

---

> ### Author Response · Authors · 2025-11-26
> **Response to Reviewer v1CK Part 1/3**
>
> We are very thankful for your positive evaluation and the recognition of the contributions and detailed assessment presented in our work. We group your comments and concerns, and respond to them as follows.
>
> > W1, Q1. Measurement-grounded input rationale: ... not directly utilizing the extracted structured measurements ... as explicit tokens or an auxiliary regression loss
>
> We thank you for raising this point about directly using structured measurements during pretraining. This is indeed an interesting design choice and we have implemented and evaluated the suggested design where the model regresses measurements during pretraining. Below we summarize what we tried and what we observed, and we will add this experiment and discussion to the revision.
>
> **Experiment design**: We first identified the 15 most frequent measurement types in EchoGround-MIMIC (listed in revision). For each measurement type we introduced a **measurement token** in the vision transformer. We concatenate 15 learned measurement tokens to the standard CLS token and image patch tokens. Each measurement token is passed through its own MLP projector to predict the corresponding scalar value. During pretraining we add an L1 regression loss on these 15 values to the CLIP objective. Due to the sparsity of our data, some measurements will be missing for certain echo views. For example, LVOT and aortic valve Doppler measurements only appear in a small subset of parasternal long-axis or apical views. We applied a masking vector to mask out measurement tokens that are absent for a given echo image. We swept `the loss ratio` between {`1.0`,`0.5`,`0.25`}. We evaluated this model on the same zero-shot disease classification and retrieval metrics used in the paper. Training and evaluation data splits remain the same as EchoVLM.
>
> **Results**: As shown below, we found no consistent improvement from this measurement regression model. For **disease zero-shot**, `best AUC` (`80.8` at $\lambda$=0.25) is only **slightly above the baseline** (`79.6`) and remains well below the EchoVLM model that uses view and negation losses (AUC 86.5 in Table 5). Precision is in fact lower for $\lambda$=1.0 and $\lambda$=0.5 than for the baseline. **For retrieval, we found overall degradation in performance compared with baseline**. Overall, the additional regression head introduces architectural complexity but does not yield consistent improvements over the baseline approach.
>
> ### Table: Effect of Measurement Regression Loss on Downstream Performance
> Best values per column are **bolded**.
>
> | Measurement Loss $\lambda$ |   AUC   | Precision | Recall |  R@1   |  R@5   |  R@10  |
> |----------------------------|:-------:|:---------:|:------:|:------:|:------:|:------:|
> | No loss                    |  79.6   | **29.0**  |  73.6  | **0.47** |  1.94  | **4.03** |
> | 1.0                        |  78.9   |   25.2    |  83.5  |  0.31  |  1.20  |  3.04  |
> | 0.5                        |  79.1   |   23.3    | **87.7** |  0.42  |  1.52  |  3.14  |
> | 0.25                       | **80.8** |   26.0    |  79.3  |  0.21  | **2.04** |  3.66  |
>
>
> **Why it doesn't improve performance**:
>
> 1. **Sparse measurements over echogram views**. Due to the highly view-dependent nature of echocardiogram, each measurement (e.g., LVIDd, AV Vmax, EF) is only available for a small subset of images acquired in specific views (e.g., PLAX, A4C, Doppler). This results in highly imbalanced supervision across the measurements. Furthermore, the effective batch size for each measurement regression task is relatively small, causing challenges to create a meaningful learning signal.
>
> 2. **Competition between losses**. Adding measurement regression increases the gradient competition between CLIP and L1 loss. The model needs to sacrifice the image-text representation quality for learning measurement values, as indicated by a drop in retrieval task.
>
> 3. **Mismatch with downstream tasks**. Our downstream tasks (e.g., disease classification, retrieval) depend on categorical labels instead of continuous measurements. Learning measurement values cannot capture the nuanced clinical semantics (e.g., the difference between EF 50% and 30% is nonlinear in diagnostic impact). Thus, the regression loss is not aligned with the model's ultimate clinical objectives. On the other hand, our measurement-grounded captions already encode these information in natural language (“biplane EF 35% consistent with moderately reduced systolic function”).
>
> Based on the above experiment, we ultimately chose not to include such measurement-regression branch in EchoVLM’s design. We will add a description of this attempted measurement-regression approach in the revision. We also highlight that using measurement regression as an auxiliary signal to improve the vision-language alignment is an interesting direction for future work. However, the sparsity of measurements in our current dataset does not adequately support this strategy at present.

---

> > ### Author Response · Authors · 2025-11-26
> > **Response to Reviewer v1CK Part 2/3**
> >
> > > W2, lack of data ablation, ... there is no experiment comparing ... a model trained on these curated "measurement-grounded captions" against a baseline model trained on the original
> >
> > We thank you for highlighting the importance of isolating the contribution of our measurement-grounded data pipeline. In response, we conducted a direct ablation in which we pretrained EchoVLM using the raw echocardiography report text from MIMIC-IV-Note as captions. We did not use additional supervision strategies such as view contrastive or text negation loss. We used the same training and evaluation split and pretraining setup as in our main experiment. As seen below, training on raw reports leads to substantially worse multimodal alignment. In **zero-shot disease classification**, **AUC drops** from `79.6` to `54.4`, and **precision drops** from `29.0` to `12.5`. We suspect the reason for such decline is that raw clinical reports contain large amounts of non-specific information that impedes image-text alignment. Therefore, we leverage OCR-extracted measurements and ASE guideline to curate both visually and clinically-grounded captions. We believe that our curated EchoGround-MIMIC provides substantial and necessary benefit for training a clinically reliable VLM.
> >
> > | Data Source       | AUC    | Precision | Recall  | Recall@5 | Recall@10 |
> > |-------------------|--------|-----------|---------|----------|-----------|
> > | Raw Reports       | 54.43 | 12.47    | 66.62  | 0.31   | 0.78    |
> > | Curated Captions  | 79.60 | 29.00    | 73.60  | 2.30   | 4.33    |
> >
> > > Q2 The paper provides a simple example for negation generation, ... could the authors clarify ...(for quantitative statements) "clinical semantic negation" takes?
> >
> > We thank you for the request for clarification. For quantitative findings, we do not negate the raw numerical value itself (e.g., “not 45%”). Instead, we map the value to its clinical interpretation and negate that interpretation, as is standard practice in clinical reporting. For example, ejection fraction is clinically used to assess systolic function, and a statement such as “Quantitative biplane left ventricular ejection fraction is 45%” indicates systolic dysfunction. The corresponding semantic negation is therefore expressed as “no systolic dysfunction was identified.” This mirrors real echocardiography reports, where negation applies to the diagnostic category rather than to the numerical measurement. We will clarify this distinction in the revised manuscript.

---

> > > ### Author Response · Authors · 2025-11-26
> > > **Response to Reviewer v1CK Part 3/3**
> > >
> > > > Q3. Considering that echocardiography is an inherently dynamic (video-based) modality ...
> > >
> > > We thank you for raising this important question. Although echocardiography is inherently dynamic, there are several practical and empirical reasons why we adopt a frame-based approach for EchoVLM.
> > >
> > > 1. **Broader and more reliable data coverage.** Real-world echo exams (including MIMIC-IV) are highly heterogeneous—many studies contain variable-length clips or only saved keyframes (e.g., ED/ES). A frame-based model can utilize all such data, whereas video models typically require complete and well-formed sequences, limiting usable volume. This is especially important in our setting because many quantitative measurements are performed on—and clinically defined by—a single representative frame rather than a full sequence.
> > >
> > > 2. **Alignment with real clinical workflows.** Cardiologists frequently base diagnostic decisions on representative frames rather than full cine loops (e.g., evaluating chamber size from a static view). A frame-level representation aligns directly with how quantitative measurements and diagnostic statements are generated in practice.
> > >
> > > 3. **Experimental evidence**. Our results show video models do not outperform frame-based VLMs. We compared EchoVLM (frame-based) against EchoPrime (video-based) to understand whether a video architecture yields advantages for the downstream tasks studied in our paper. The results are presented as follows.
> > >
> > > 3.1 Disease Classification and Retrieval.
> > >
> > > We adapted EchoPrime to our image–text setting by converting each image into a pseudo-video (3×16×224×224) and evaluating it under the exact same data splits on EchoGround-MIMIC as with other VLMs.
> > >
> > > As shown below, EchoPrime performed worse than EchoVLM in both disease zero-shot classification and image–text retrieval: disease AUC = 82.8 (vs. 86.5 for EchoVLM), and retrieval recall-5 = 1.4% (vs. 3.0% for EchoVLM). These results confirm that EchoPrime is not optimized for image–text alignment or retrieval tasks, consistent with its design as a video-level report aggregation model rather than a CLIP-style VLM. We note that training EchoPrime incurred **16 times** higher memory requirements compared with our image-based EchoVLM, so we did not use it as the main comparison baseline.
> > >
> > > | Model      | AUC      | Precision | Recall     | Recall@5    | Recall@10   |
> > > |------------|----------|-----------|------------|-------------|-------------|
> > > | EchoPrime  | 82.8 | 29.7  | 93.8   | 1.41    | 2.78    |
> > > | EchoVLM (ours)    | 86.5 | 34.2  | 86.2   | 2.98    | 5.70    |
> > >
> > > 3.2 Vision Task Evaluation (Fine-tuning)
> > >
> > > We further evaluated EchoPrime’s vision encoder on the downstream task of view classification. We considered two settings: 1) EchoPrime-FT-Sequence: fine-tuned with 16 real consecutive frames; 2) EchoPrime-FT-Static: fine-tuned with the same frame repeated 16× (to match same information received by image-based models). The results are as follows. Even with sequential frames, EchoPrime remains below EchoVLM.
> > >
> > > |                       |  F1  | Precision | Recall |
> > > |:---------------------:|:----:|:---------:|--------|
> > > | EchoPrime-FT-Sequence | 93.8 |    94.9   | 93.2   |
> > > | EchoPrime-FT-Static   | 92.2 |    92.5   | 92.1   |
> > > | EchoVLM (ours)        | 95.3 |    95.8   | 95.1   |
> > >
> > > 3.3 k-NN Evaluation (No Fine-Tuning)
> > >
> > > To avoid confounding effects (e.g. hyperparameter tuning) from fine-tuning, we also followed the DINO's practice (code and parameter default) and performed K-NN evaluation on both internal and public (unseen) dataset: freezing the vision backbone and classify the test dataset with k=20 nearest neighbors in training dataset for both EchoPrime and EchoVLM.
> > >
> > > | Model          | Dataset      | F1   | Precision | Recall |
> > > |----------------|--------------|------|-----------|--------|
> > > | EchoPrime      | Internal-26k | 38.9 | 38.7      | 53.1   |
> > > | EchoVLM (ours) | Internal-26k | 53.3 | 54.2      | 62.7   |
> > > | EchoPrime      | CAMUS-1k     | 79.3 | 80.0      | 84.5   |
> > > | EchoVLM (ours) | CAMUS-1k     | 94.0 | 94.0      | 94.0   |
> > >
> > > On CAMUS (A2C/A4C views), the confusion matrices of 100 videos in testing dataset shows EchoVLM’s stronger vision representation:
> > >
> > > | EchoPrime | A2C | A4C |   | EchoVLM (ours) | A2C | A4C |
> > > |:---------:|:---:|:---:|:-:|:--------------:|:---:|:---:|
> > > |    A2C    |  49 |  19 |   |       A2C      |  47 |  3  |
> > > |    A4C    |  1  |  31 |   |       A4C      |  3  |  47 |
> > >
> > > Across classification, retrieval, view recognition, and k-NN evaluations, (video-based) EchoPrime underperforms (frame-based) EchoVLM.
> > >
> > > We emphasize that our use of a frame-based model is not intended as the final solution for echo. Given the heterogeneous and often incomplete nature of real-world echo studies, we believe a hybrid frame- and video-based framework is ultimately more clinically appropriate, which will be a focus of future work. Thank you for the constructive feedback.

---

### Official Review · Reviewer_Frhu · 2025-10-29

**Soundness:** 3
**Presentation:** 3
**Contribution:** 3
**Rating:** 6
**Confidence:** 5

**Summary:**

The paper introduces the EchoGround-MIMIC dataset, a set of image/text paired datasets for echocardiography. Specifically, the authors use the MIMIC-IV-ECHO  dataset and extract numerical measurements using OCR-based methods. Second, the authors propose a CLIP-based contrastive learning framework and evaluate EchoVLM on 5 different clinical applications with 36 clinical tasks.

**Strengths:**

- The authors propose a needed dataset for echocardiography. Most of the papers working on VLMs for echo are constrained to private datasets, limiting their applicability and contribution.

- The paper comprehensively details different procedures taken to obtain the final dataset from the raw original MIMIC-IV-ECHO.

- The negation-aware contrastive objective for CLIP, along with diverse ablation studies.

**Weaknesses:**

- The main weakness of the paper, to me, is its limited architectural novelty. Although introducing the new dataset is needed for the community working on echocardiography, the proposed Echo-VLM is similar to prior works originally CLIP and also its variants Echo-CLIP.

- Measurements are cropped from overlays and transcribed via an LLM, along with the captions and guideline labels. Although this is acknowledged in the paper and despite manual checks, parsing errors may introduce label noise as mentioned. To what extent is this labelling noise mitigating? Were there cardiologists involved in the process?

-

**Questions:**

- Can the authors elaborate on their novelty in terms of the architecture design, as opposed to prior works like EchoCLIP?

- A main concern of mine is whether the dataset is really going to be open-sourced. I understand that the authors mention this; however, based on my experience in this field, I have seen many papers in top-tier conferences that mention they will open-source the code/data, but they don't. This is particularly evident in many echo papers. Ideally, the authors could share an anonymised GitHub repo containing the code/data of the paper.

- Can the authors clarify more on the manual checks performed on the outputs of LLMs? How trustful is the outputs of the OCR algorithtm and the LLM-generated captions?

**Details Of Ethics Concerns:**

N/A.

---

> ### Author Response · Authors · 2025-11-26
> **Response to Reviewer Frhu Part 1/2**
>
> We sincerely appreciate your positive evaluation and the recognition of the contributions and detailed methodology presented in our work. We group your comments and concerns, and respond to them as follows.
>
> > W1, Q1: Limited architectural novelty, compare to EchoCLIP
>
> We thank you for your comment regarding architectural novelty. Our goal in EchoVLM is not to redesign the CLIP architecture, but to introduce clinically meaningful multimodal pretraining objectives that address limitations of prior VLMs.
>
> First, we make the following comparison with EchoCLIP. EchoCLIP trains on entire clinical reports, which contain long, noisy, and often view-irrelevant text. As shown in our data curation ablation study, we found that **directly aligning echogram frames with reports lead to poor disease classification and retrieval performance**. Alternatively, EchoVLM uses measurement-grounded captions that explicitly reference OCR-extracted measurements (e.g., EF, LV dimensions, valve gradients) and ASE guideline criteria. This provides structured, quantitative supervision that EchoCLIP does not capture. Our data ablation confirms this is essential: training on raw reports yields AUC 54.0, whereas curated measurement-grounded captions yield AUC 80.0.
>
> Second, EchoVLM proposes **two new clinically meaningful pretraining objectives tailored to echocardiography**. We first introduce view-informed contrastive loss. EchoVLM explicitly incorporates the ASE-defined view hierarchy into the contrastive objective. Images from the same view are treated as positives, and images from different views as structured negatives. EchoCLIP does not use any view-aware learning objective, despite the fact that anatomical appearance in echocardiography is fundamentally view-dependent. This structured view supervision improves vision tasks (e.g., +1.1% improvement in view classification accuracy). We also propose negation-aware contrastive learning. Clinical echo interpretation relies heavily on negations (e.g., “no effusion”, “no significant regurgitation”). EchoCLIP treats negations as ordinary text and provides no mechanism to differentiate between affirmative vs. negated findings. EchoVLM introduces a negation-aware loss using LLM-generated counterfactual captions that contrast affirmative and negated clinical statements. This yields considerable improvements in zero-shot disease detection (e.g., +2.3% AUC, +2.3% recall).
>
> Although the underlying architecture remains CLIP-like, **the training objectives and supervision signals are new and clinically grounded, capturing quantitative measurement reasoning and view-dependent anatomical structure**. None of these supervision mechanisms are present in EchoCLIP or CLIP. Collectively, these components enable EchoVLM to outperform all prior vision–language and vision-only foundation models across multimodal tasks and vision tasks. Thus, EchoVLM’s novelty lies not in architectural changes but in introducing clinically structured multimodal learning objectives that better reflect the real workflow of echocardiography.
>
> > Q2:  main concern of mine is whether the dataset is really going to be open-sourced...
>
> We completely understand this concern. Working in this field, we have also experienced the similar challenges. Please note that per the license terms of the MIMIC-IV data collection (https://physionet.org/content/mimic-iv-echo/view-dua/0.1/, Point 9), which states: “If I openly disseminate my results, I will also contribute the code used to produce those results to a repository that is open to the research community.” We are committed to contributing our derived data back to the public community.
>
> To support reproducibility, we have prepared an anonymized GitHub repository (https://anonymous.4open.science/r/EchoVLM-9CE4) that includes:
>
> - all code for OCR and LLM-based report processing
> - all code for model training and evaluation
> - structured output including both intermediate and final results (JSON files containing paths to DICOMs, OCR results, extracted measurements, and generated captions)
>
> Because of time constraints, the initial README is concise, but we are continuing to expand it. Finally, we emphasize that **access to the underlying DICOM images cannot be redistributed by us**. Per the MIMIC-IV license (Point 7), obtaining the raw imaging data requires completing the official CITI/HIPAA training and applying through PhysioNet. This requirement is mandatory, and we cannot bypass it. We thank you for your understanding.

---

> > ### Author Response · Authors · 2025-11-26
> > **Response to Reviewer Frhu Part 2/2**
> >
> > > W1, Q3:  Measurements are cropped from overlays and transcribed via an LLM... Can the authors clarify more on the manual checks performed on the outputs of LLMs, how trustful is the outputs of the OCR ..
> >
> > **OCR Measurement Extraction and Cleaning.** The OCR step is used exclusively to extract measurement name–value pairs. After the first pass, we found `1,232 unique keys`—many of which represented the same measurement with slightly different naming conventions (e.g., AV_Vmax vs. AV Vmax), while others corresponded to non-clinical overlays such as “gain,” “velocity scale,” or “time scale.”
> > We focused on key–value pairs appearing more than 10 times, resulting in `278 candidates`. Based on clinical practice, all measurements should fall under `11 categories` (LV, LA, RV, RA, MV, TV, AV, PV, SV, PVein, Aorta). Our team manually reviewed these categories and cleaned/classified the 278 candidates into `167 final structured measurements`, which are released in our repository organized in pydantic classes (see `data/mimic_anatomy.py` in shared repo).
> >
> > This manual cleanup significantly improved downstream reliability. After consolidation, we observed that the LLM almost never mis-parsed the cleaned measurement strings into structured dictionaries. The final dataset includes structured metadata for each measurement (anatomy, name, value, unit, cardiac phase). We are also releasing the intermediate cleaning results for full transparency.
> >
> > **Trustworthiness of LLM-Generated Captions.** We collaborated closely with a cardiologist to guide the measurement curation and establish normal ranges for relevant quantitative measurements. To validate the LLM-generated disease captions, we implemented a rule-based consistency check: for each disease, and for each measurement associated with that disease, we compared the LLM’s disease label against a rule-based label derived from the numerical value.
> >
> > We found strong alignment in the majority of cases—approximately `87%`, where values were clearly within normal or abnormal ranges. The remaining `~13%` of cases involved _borderline values_ or clinically _subjective_ thresholds. For example, although the normal LA length is typically 3.5–5.2 cm, an LA length of 4.9 cm may still be described as “dilated” in the clinical report due to patient-specific factors such as age, comorbidities, or longitudinal trends unavailable in isolated measurements. In such cases, we retain the report-derived label but mark these samples with a binary "matched_with_LLM" flag in our JSON dataset to allow users to re-filter or re-interpret them as needed.
> >
> > During manual verification, we also removed diseases related to the mitral valve, mitral regurgitation, and stroke volume where measurement–caption inconsistencies were most prominent. They also rely on multi-parameter Doppler criteria and hemodynamic context that are not available in our context. To ensure dataset quality and reduce clinically implausible labels, we exclude these categories from the measurement-grounded disease labels.
> >
> > We sincerely thank you again for raising these valuable points and hope the added details help strengthen the clarification. We will incorporate these explanations into the revised manuscript.

---

### Official Review · Reviewer_zREj · 2025-10-30

**Soundness:** 3
**Presentation:** 2
**Contribution:** 3
**Rating:** 6
**Confidence:** 4

**Summary:**

The paper presents a vision-language model, CLIP-style, for echocardiography. VLMs for echocardiography suffer from internal challenges in accurate measurement predictions, and sparse and unfocused data of image-text pairs. The proposed model combines image and text encoders trained on the novel EchoGround-MIMIC dataset (19,065 measurement-grounded image-text pairs). The dataset comes from preprocessing and organizing existing public repos (MIMIC-ECHO), and could be valuable for further research—so far there is no similar open-source data, therefore the data is useful. The model uses two specialized contrastive losses: a view-informed contrastive loss (same-view positives, different-view negatives) and a negation-aware contrastive loss for distinguishing negative vs. positive clinical findings in text. However, these losses appear to offer limited technical novelty. For downstream tasks like segmentation and landmark detection, task-specific heads are added to the pre-trained encoder and fine-tuned on benchmark datasets.

**Strengths:**

-- Data: EchoGround-MIMIC (~20K measurement-grounded image-text pairs) - first open-source dataset of its kind for echocardiography. Data Processing Innovation -- Successfully integrates MIMIC-IV-ECHO imaging with MIMIC-IV-Note reports.


-- Clinical Relevance: Addresses critical gap between free-text narratives and quantitative measurements essential for guideline-based echo diagnosis.

-- Comprehensive Evaluation Framework: 36 tasks across 5 clinical application types (classification, retrieval, segmentation, landmark detection) - would be valuable if released as a benchmark.

-- Community Value: Fills significant resource gap for medical AI research in echocardiography.

**Weaknesses:**

-- Limited Technical Novelty: View-informed loss is just constrained negative sampling; negation-aware loss potentially similar to existing work (e.g., MICCAI 2025, "EchoViewCLIP: Advancing Video Quality Control through High-performance View Recognition of Echocardiography")

-- Evaluation Methodology Issues: Primary comparison against unreleased EchoApex (weights/data are not released, based on reported results) instead of available EchoPrime (weights are open to download) raises reproducibility concerns, and if all models were trained in the same manner.

-- Technical Details: Frame vs. video level processing unclear; mathematical formulation of negation-aware loss may lack sufficient innovation

-- Algorithmic Contributions Questionable: Technical contributions may not meet novelty bar for top-tier venues - relies heavily on dataset contribution rather than methodological innovation

**Questions:**

Given that EchoPrime is publicly available while EchoApex is unreleased, why not use EchoPrime as the primary baseline? This would enable reproducible comparisons and address potential selection bias concerns.

---

> ### Author Response · Authors · 2025-11-26
> **Response to Reviewer zREj Part 1/2**
>
> We are very thankful your detailed assessment and are encouraged by the recognition of our dataset contribution, clinical relevance, and broad evaluation. As the comments center on (1) reproducible benchmarking and (2) methodological novelty, we summarize our responses to these points below.
>
> > Q1, W2. Given that EchoPrime is public while EchoApex is not, why not use EchoPrime as the primary baseline? Evaluation ... raises reproducibility questions.
>
> We agree that reproducible baselines are important, and for this reason we made substantial effort to adapt EchoPrime into a VLM for a direct comparison. To fully assess its performance, we conducted three complementary studies: (1) disease classification and image–text retrieval, (2) fine-tuned view classification, and (3) k-NN evaluation without fine-tuning on both internal and public datasets.
>
> **1.1 Disease Classification**
>
> Because EchoPrime is designed for study-level video modeling, we need to adapt its data processing for our image-text contrastive setting. Specifically, we initialized the model from the released checkpoints, and converted each image into a pseudo-video by concatenating frames into a 3 × 16 × 224 × 224 input. We then trained and evaluated EchoPrime on the same EchoGround-MIMIC splits and protocols used for all other VLMs.
>
> As shown below, EchoPrime performed worse than EchoVLM in both disease zero-shot classification and image–text retrieval: disease AUC = 82.8 (vs. 86.5 for EchoVLM), and retrieval recall-5 = 1.4\% (vs. 3.0\% for EchoVLM). These results confirm that EchoPrime is not optimized for image–text alignment or retrieval tasks, consistent with its design as a video-level report aggregation model rather than a CLIP-style VLM.
> We note that training EchoPrime incurred **16 times** higher memory requirements compared with our image-based EchoVLM, so we did not use it as the main comparison baseline.
>
>
> | Model      | AUC      | Precision | Recall     | Recall@5    | Recall@10   |
> |------------|----------|-----------|------------|-------------|-------------|
> | EchoPrime  | 82.8 | 29.7  | 93.8   | 1.41    | 2.78    |
> | EchoVLM (ours)    | 86.5 | 34.2  | 86.2   | 2.98    | 5.70    |
>
>
> **1.2 Vision Task Evaluation (Fine-tuning)**
>
> We further evaluated EchoPrime’s vision encoder on the downstream task of view classification. We considered two settings: 1) EchoPrime-FT-Sequence: fine-tuned with 16 real consecutive frames; 2) EchoPrime-FT-Static: fine-tuned with the same frame repeated 16× (to match same information received by image-based models). The results are as follows. Even with sequential frames, EchoPrime remains below EchoVLM.
>
>
> |                       |  F1  | Precision | Recall |
> |:---------------------:|:----:|:---------:|--------|
> | EchoPrime-FT-Sequence | 93.8 |    94.9   | 93.2   |
> | EchoPrime-FT-Static   | 92.2 |    92.5   | 92.1   |
> | EchoVLM (ours)        | 95.3 |    95.8   | 95.1   |
>
>
> **1.3 k-NN Evaluation (No Fine-Tuning)**
>
> To avoid confounding effects (e.g. hyperparameter tuning) from fine-tuning, we also followed the DINO's practice (code and parameter default) and performed K-NN evaluation on both internal and public (unseen) dataset: freezing the vision backbone and classify the test dataset with k=20 nearest neighbors in training dataset for both EchoPrime and EchoVLM.
>
>
> | Model          | Dataset      | F1   | Precision | Recall |
> |----------------|--------------|------|-----------|--------|
> | EchoPrime      | Internal-26k | 38.9 | 38.7      | 53.1   |
> | EchoVLM (ours) | Internal-26k | 53.3 | 54.2      | 62.7   |
> | EchoPrime      | CAMUS-1k     | 79.3 | 80.0      | 84.5   |
> | EchoVLM (ours) | CAMUS-1k     | 94.0 | 94.0      | 94.0   |
>
> On CAMUS (A2C/A4C views), the confusion matrices of 100 videos in testing dataset shows EchoVLM’s stronger vision representation:
> | EchoPrime | A2C | A4C |   | EchoVLM (ours) | A2C | A4C |
> |:---------:|:---:|:---:|:-:|:--------------:|:---:|:---:|
> |    A2C    |  49 |  19 |   |       A2C      |  47 |  3  |
> |    A4C    |  1  |  31 |   |       A4C      |  3  |  47 |
>
> Across all evaluations, EchoPrime underperforms EchoVLM. Our results suggest that EchoPrime’s architecture is strongly optimized for video–language alignment, and without specific visual modeling components, strong vision performance is difficult to achieve.

---

> ### Author Response · Authors · 2025-11-26
> **Response to Reviewer zREj Part 2/2**
>
> > W3. Frame vs. video-level design unclear
>
> We thank you for raising this important question. There are several practical and empirical reasons why EchoVLM adopts a frame-based design.
>
> **Broader and more reliable data coverage.** Real-world echocardiography exams (including MIMIC-IV) are highly heterogeneous—many studies contain variable-length clips, missing frames, or only saved keyframes (e.g., ED/ES). A frame-based model can utilize all such data, whereas video
> models typically require complete and well-formed sequences, which greatly limits usable volume. This is especially important in our setting because many quantitative measurements are performed on—and clinically defined by—a single representative frame rather than a full sequence.
>
> **Alignment with real clinical workflows.** Cardiologists frequently base diagnostic decisions on representative frames rather than full cine loops (e.g., evaluating chamber size from a static view). A frame-level representation aligns directly with how quantitative measurements and diagnostic  statements are generated in practice. Experimental evidence: video models do not outperform frame-based VLMs
>
> We also note that our use of a frame-based model in this work is not intended as the final design for echocardiography. Because echo imaging is inherently dynamic, many tasks naturally benefit from temporal modeling. However, given the heterogeneous and often incomplete nature of real-world echo exams, we believe that a hybrid framework—integrating both frame-based and video-based representations—will ultimately offer the most robust and clinically aligned solution.
>
> > W1. W3, W4. Limited technical novelty..  with similar works (e.g., EchoViewCLIP), negation-aware loss, and algorithmic contributions
>
> We thank you for this comment. While our contrastive losses build upon established principles, their purpose is to introduce clinically meaningful inductive biases, specifically view structure and negation semantics that are not captured by existing VLMs. Our view-informed loss is not "just constrained negative sampling". In echocardiography, multiple frames can share the same ASE view (A4C, PLAX, etc.), so standard CLIP loss cannot capture this underlying structure. View-informed loss is implemented via a view-conditioned positive mask over the batch, not just by dropping a subset of negatives. Importantly, we show that it improves not only view classification but also zero-shot disease classification and transfer to downstream segmentation and landmark detection.
>
> Regarding the negation-aware loss, we argue that this loss is a simple and effective approach to enable medical VLMs to understand clinical negations, since positive prompts and negative prompts have many token overlaps. We will also make our text dataset with LLM negated captions available to advance research in this area.
>
> We pointing out differences to recent EchoViewCLIP as follows.
>
> 1. **Fundamental differences in data curation goals and scope.** EchoViewCLIP is designed for view recognition and OOD detection. Its “negation” mechanism simply negates view labels (e.g., “not PLAX”), and does not address the important distinction between fine-grained disease states such as mild mitral regurgitation vs. no significant mitral regurgitation. In contrast, our EchoVLM is clinically grounded by measurements-based disease captions from reports. We then use an LLM to create counterfactual disease captions, ensuring negations that follow ASE guidelines and patient labels. Thus, our negation-aware loss operates on clinically meaningful disease semantics, which is essential for downstream zero-shot disease classification.
>
> 2. **Simpler architecture.** EchoViewCLIP introduces two visual experts and an Negation Semantic-Enhanced Detector to reject OOD views. Our EchoVLM follows a simple joint encoder design like CLIP. We do not introduce additional encoders, visual experts, or OOD-specific branches for multi-modal learning.
>
> 3. **Clinically relevant evaluations.** EchoViewCLIP only focuses on view recognition, whereas EchoVLM has been extensively tested for disease zeroshot classification, retrieval, view classification, segmentation, landmark detection across 36 tasks. This shows that our measurement-grounded captions, view-informed contrastive objective, and negation-aware learning yield a unified representation that generalizes across tasks and datasets for echocardiogram learning.
>
> We thank you for the positive and encouraging comments on our strengths and contributions again. We have made an anonymous repository (https://anonymous.4open.science/r/EchoVLM-9CE4) to provide early access to the curated dataset with the aim to contribute to the community. We sincerely appreciate your feedbacks and we will integrate the aforementioned findings and clarifications into the revised manuscript.

---

### Author Response · Authors · 2025-11-26
**General response**

We sincerely thank all four reviewers for their constructive feedback. We are encouraged that reviewers consistently recognized the **importance of our measurement-grounded multimodal dataset** (`zREj`, `Frhu`, `v1CK`, `qUxR`), **the clinical relevance of our objectives**, and **comprehensive evaluation across 36 tasks** (`zREj`, `Frhu`, `v1CK`). Reviewers highlighted the value of introducing clinically motivated contrastive learning that captures view structure and negation semantics (`Frhu`, `v1CK`, `qUxR`) and the community impact of providing a high-quality, reproducible, open dataset (`zREj`, `Frhu`).

We have carefully reviewed **all** concerns and added substantial revisions and experiments. Below is a summary of the key contributions and updates; detailed point-by-point responses follow.

## Main contributions
---
**Measurement-grounded multimodal dataset for echocardiography**

Reviewers agreed that EchoGround-MIMIC fills a critical gap as the first measurement-grounded, guideline-aligned echo dataset, pairing images with standardized views, structured OCR-extracted measurements, measurement-grounded captions, and ASE-derived disease labels (`zREj`, `Frhu`, `v1CK`, `qUxR`).

---
**Clinically informed multimodal learning objectives**

Reviewers noted the importance of our view-informed contrastive loss (capturing ASE-standard view structure) and negation-aware contrastive loss (distinguishing clinically negative vs. affirmative findings). (`Frhu`, `v1CK`)

---
**Extensive evaluation demonstrating strong transferability**

Reviewers highlighted our evaluation—disease classification, retrieval, view classification, segmentation, landmark detection and noted that EchoVLM consistently outperforms prior VLMs and matches or surpasses vision foundation models on key tasks (`zREj`, `v1CK`, `qUxR`). The generalization across multiple public datasets and clinical sites was also well received.

## Response and Update Summary
---
In direct response to reviewer requests, the revised manuscript incorporates the following updates (all highlighted in **blue**):

### Major updates addressing concerns shared by multiple reviewers

1. **Expanded comparison with EchoPrime**, including disease classification, image–text retrieval, vision fine-tuning, and k-NN evaluations (**Table 1, Table 2, Appendix A.8, Figure 11**) to address reproducibility and baseline fairness concerns (`zREj`, `v1CK`, `qUxR`). Our proposed EchoVLM achieves improved performance over EchoPrime in all these experiments.

2. **Detailed data quality and derived dataset verification.** We provide comprehensive descriptions of our OCR pipeline, structured-measurement extraction, manual checks on key fields, and even (representative) intermediate examples. We also include a baseline experiment using raw clinical reports directly as language supervision
(**Appendix A.5**) to address data-quality and label-noise concerns (`Frhu`, `qUxR`).

3. **Release of code and dataset.** To support reproducibility, we have released our processed datasets, OCR outputs, structured measurements, raw measurement keys, training/evaluation code, along with instructions for regenerating intermediate datasets. The repository is currently anonymous and uploaded here: https://anonymous.4open.science/r/EchoVLM-9CE4

### Additional experiments and evaluations

4. **Varied pretraining methodology**: measurement-regression based pretraining comparison (`v1CK`), we clarified why explicit regression is not optimal.

5.  **Ablation on data curation**, directly comparing raw reports vs. our measurement-grounded captions (**Section 4.3**) to address questions about the necessity and impact of measurement-grounded captions (`v1CK`, `qUxR`).

6. **Analysis of generalization across institutions**, including multi-site U.S., Asian, and European (CAMUS) results
(**Appendix A.7**) to address external-validity concerns (`qUxR`).

7. **Inference efficiency profiling for real clinical workflows**, reporting latency and memory use across downstream tasks
(**Section 4.5**) to address deployment concerns (`qUxR`).


### Additional statistical and qualitative analysis

8. Expanded description and examples of semantic negation, especially for quantitative findings (**Introduction**) to address interpretability concerns (`v1CK`, `zREj`).

9.  Expanded explanation of the view-informed loss and its relationship to CLIP (`Frhu`, `qUxR`).

10.  Failure-mode analysis for disease classification (LV false positives) (`qUxR`).

11. Additional clarifications on frame-based vs. video-based design, supported by new EchoPrime experiments (`v1CK`, `qUxR`).

Beyond these targeted updates, we are continuing to refine clarity and organization throughout the manuscript. We sincerely thank the reviewers again for their insightful feedback, which has substantially improved the quality and transparency of this work.

---

### Note · Authors · 2026-01-26

I have read and agree with the venue's withdrawal policy on behalf of myself and my co-authors.

---

### Meta-Review · Area_Chair_LXJw · 2026-01-23

**Summary:**

While the reviewers agree that the paper presents an important dataset, there are several weaknesses mentioned by the reviewers. Lack of technical novelty over CLIP’s contrastive loss, and view-invariance contrastive loss. The models operating on single frames when EchoCardiography is inherently operational on video modality. Lack of diversity in data coming from a single institution. A disconnect between the “measurement grounded” narrative and the methodology and results.

**Reviewer Concerns:**

Technical Novelty: I agree with the reviewers that the proposed methodology is fairly limited for ICLR. The proposed loss term has several elements from existing works like CLIP (language-image pretraining), SLIP (image self-supervision), and text-encoder self-supervision with negation (con-clip, NegCLIP). While these losses have been applied for the first time to echocardiographic data domains, that would not be considered technically novel.

While it is great that the authors compare an echoPrime on their benchmarks, the results should definitely be added to the paper. To claim this as “experimental evidence” for not using video as a modality instead of images does not seem fair. It is entirely possible that on the same type of dataset, a model using video modality does better than frames. A fair comparison would have been to test the models on a video dataset, say from Echoprime, and show that this frame-based model is strictly better than video models on video datasets as well.

The authors have conclusively shown that even though the data source is singular, it still has generalization abilities.

I agree with reviewer v1CK that there is a gap in methodology and narrative in relation to the date being "measurement-grounded". The authors could have softened their claims if the captions indeed only summarize such measurements. Instead the rebuttal shows results, that such measurement supervision actively hurts model performance. The result does not show that the original model built on dataset
is "measurement-grounded" and instead shows that being "measurement-grounded" is actually not a useful property for such a VLM on downstream tasks.

**Reviewer Scores:**

All the reviewers' scores are unlikely to increase.

The two critical reviewers' scores will likely not increase, as they have a fundamental issue with the technical novelty of the papers.

The two positive reviewers also did not seem too positive about the work and raised several reasonable weaknesses that are still unaddressed. Therefore the scores are likely to decrease.

---

### Decision · Program_Chairs · 2026-01-26

Reject